# Structural Optimization and MEMS Implementation of the NV Center Phonon Piezoelectric Device

**DOI:** 10.3390/mi13101628

**Published:** 2022-09-28

**Authors:** Xiang Shen, Liye Zhao, Fei Ge

**Affiliations:** Key Laboratory of Micro-Inertial Instrument and Advanced Navigation Technology, Ministry of Education, School of Instrument Science and Engineering, Southeast University, Nanjing 210096, China

**Keywords:** nitrogen-vacancy center, phonon piezoelectric device, phonon-coupled manipulation, acoustic characteristics, structural optimization, MEMS implement

## Abstract

The nitrogen-vacancy (NV) center of the diamond has attracted widespread attention because of its high sensitivity in quantum precision measurement. The phonon piezoelectric device of the NV center is designed on the basis of the phonon-coupled regulation mechanism. The propagation characteristics and acoustic wave excitation modes of the phonon piezoelectric device are analyzed. In order to improve the performance of phonon-coupled manipulation, the influence of the structural parameters of the diamond substrate and the ZnO piezoelectric layer on the phonon propagation characteristics are analyzed. The structure of the phonon piezoelectric device of the NV center is optimized, and its Micro-Electro-Mechanical System (MEMS) implementation and characterization are carried out. Research results show that the phonon resonance manipulation method can effectively increase the NV center’s spin transition probability using the MEMS phonon piezoelectric device prepared in this paper, improving the quantum spin manipulation efficiency.

## 1. Introduction

The nitrogen-vacancy (NV) center is an atomic defect in diamonds. It has received extensive attention since the development of optical detection magnetic resonance technology [1,2]. The solid-state single-spin system of the NV center has a coherence time of milliseconds. It can be initialized and read out by the optical detection magnetic resonance (ODMR) and can be controlled by the alternating magnetic field. These excellent properties make it applicable in quantum computing [3], quantum simulation [4], and quantum physics [5,6,7,8]. It has broad application prospects in physics, chemistry, and biology.

Coupling the phonon field to qubits has sparked a research boom in quantum acoustics. Phonons are vibrating quantized representations of ions or atoms in crystals [9]. Phonon characteristics are closely related to the transport properties of crystalline materials, elementary excitation lifetimes, spectral line widths, magnetic susceptibility, and thermal properties. Phonons have been applied to superconducting qubit computing [10,11,12,13,14,15,16] because of the advantages of small wavelength and no radiation. The spin entanglement, spin squeezing, and phonon transduction have been carried out by coupling phonon to artificial dopants mechanically or electromagnetically, such as the NV center [17,18,19]. Scholars have found that the spin state of the NV center is related to the phonon mode of a diamond [20,21,22,23,24,25,26,27,28,29,30,31]. The deformation potential of the phonon–spin coupling strength of the NV center is related to the energy gap, and its spin coherence leads to a weak phonon coupling of the ground state spin. The degree of phonon coupling in the excited state spin is six orders of magnitude larger than in the ground state spin [32,33]. Therefore, the coupled control of the NV center can be achieved through phonon-coupled multiphysics [34].

The phonon piezoelectric device of the NV center is a mechanical and electrical coupling device that can provide phonon-coupled manipulation. Mathematical modeling, finite element method, and equivalent circuit modeling are used in designing and optimizing the phonon device structure of the NV center [35]. Diamond is a semi-infinite substrate, and ZnO is the piezoelectric layer in the phonon piezoelectric device. The phonon propagation characteristics can be analyzed intuitively using the finite element method. It depends on the structural and material parameters of the phonon piezoelectric device. In order to verify the phonon-coupled spin manipulation method, the structure of the phonon device of the NV center is designed and optimized, and the structure is a MEMS, implemented and experimentally verified in this paper.

The paper is organized as follows: In Section 2, a phonon-coupled manipulation mechanism and acoustic characteristics of the phonon piezoelectric device of the NV center are presented. In Section 3, the structural optimization of phonon piezoelectric devices of the NV center is described. Phonon-coupled manipulation experiment of the NV center is reported in Section 4. Finally, conclusions are presented in Section 5.

## 2. Phonon-Coupled Manipulation Mechanism and Acoustic Characteristics of the Phonon Piezoelectric Device of the NV Center

### 2.1. Phonon Resonance Structure and Acoustic Manipulation Mechanism of the NV Center

A negatively charged NV center is selected in this paper, which contains six electrons, two from nitrogen atoms, three from carbon atoms adjacent to the vacancy, and one trapped electron. The structural model of the phonon piezoelectric device of the NV center design is shown in Figure 1a, including an interdigital transducer (IDT) layer, a ZnO piezoelectric membrane layer, and a diamond substrate. The ZnO piezoelectric membrane is grown on the surface of the diamond. IDT is deposited on the surface of the ZnO piezoelectric membrane layer with the phonon-coupled manipulation method, multiphysics fields such as magnetic field, radiofrequency (RF) field, microwave, and laser perform resonant coupling regulation of the NV center. Phonon-coupled manipulation mechanism of the NV center is shown in Figure 1b. The blue sideband transition is manipulated by the phonon coupling of the phonon field-driven system from |g> state to |e> state in the presence of a phonon-containing field. This system is equivalent to a spin transition between two lower state phonon ladders, where $\Omega_{P}$ is the phonon equivalent Larmor frequency, $\Omega_{0}$ is the actual Larmor frequency, and $\Omega_{1}=\Omega_{P}+\Omega_{0}$ is the effective Larmor frequency. The frequency of the laser is in a normal distribution interval (as shown in the green area in Figure 1b). For the non-phonon-coupled case, photons with wavelengths of 532 nm (whose corresponding actual Larmor frequency is $\Omega_{0}$) can achieve excitation of the NV center. For the phonon-coupled case, photons with a wavelength of more than 532 nm (whose corresponding actual Larmor frequency is $\Omega_{0}=\Omega_{1}-\Omega_{p}$) can achieve excitation of the NV center. More photons transition from the ground state to the excited state for the phonon-coupled case.

The excited states of the NV center are strongly coupled to the wavelength lattice strain. This electron–phonon coupling can lead to strain-induced energy transfer. For the $E_{y}$ states, the electron–phonon interaction Hamiltonian model describing strain-induced energy transfer is [34]:(1)$$H_{\text{e-phonon}}=\hbar g_{2}\left({b+b}^{+}\right)\left|E_{y}{><E}_{y}\right|$$
where b is the annihilation operator of phonon, $b^{+}$ is the generation operator of phonon, $g_{2}{=\delta \varepsilon }_{c}k_{m}\sqrt{\hbar {/2\;m\omega }_{m}}$ is the effective coupling rate of electron–phonon, $\varepsilon_{c}$ represents the band edge energy of lattice vibration, ${\delta \varepsilon }_{c}$ is the deformation potential, ${\delta \varepsilon }_{c}=\left(\frac{\partial \varepsilon_{c}}{\partial V}\right)\delta V$, δV is the local change in lattice volume due to longitudinal acoustic modes of lattice vibrations, $k_{m}$ is the wave number of the phonon mode, m is the effective mechanical mass oscillator, $\omega_{m}$ is the phonon oscillation frequency, and $E_{y}$ is the strain-induced energy transfer excited state.

For spin transitions manipulated by phonon coupling, from $m_{s}=0$ to $E_{y}$ state, the effective Hamiltonian $H_{R}$ is [34]:(2)$$H_{R}=\hbar \cdot \frac{\Omega_{0}}{2}\cdot \frac{g_{2}}{\omega_{m}}\left(b\left|E_{y}{><m}_{s}=0\right|+b^{+}\left|m_{s}{=0><E}_{y}\right|\right)$$
where $\Omega_{0}$ is the Larmor frequency coupled to the state transition. The effective Larmor frequency of the spin transition manipulated by phonon coupling is:(3)$$\Omega_{P}{=g}_{2}\sqrt{\langle n\rangle }\cdot \Omega_{0}{/\omega }_{m}$$
where 〈n〉 is the average number of phonons.

Spin transfers into a three-level system during the phonon-coupled manipulation. We use electron-phonon coupling to mediate the interaction between the NV center spins and mechanical degrees of freedom. Electrons are trapped in two lower states, forming a dark state resulting from the coherent superposition, which mediates and controls the interaction between the NV center spin states and phonon modes, given by:(4)$${|\psi }_{d}>=\frac{1}{\sqrt{\Omega_{P}^{2}{+\Omega }_{\pm }^{2}}}{(\Omega }_{P}{|m}_{s}{=\pm 1>-\Omega }_{\pm }{|m}_{s}=0)$$
where $\Omega_{P}$ and $\Omega_{\pm }$ are the Rabi frequencies of two transitions coupled to $m_{s}=0$ and $m_{s}=\pm 1$. Therefore, its Hamiltonian model for the quantum measurement of the NV center for the phonon-coupled manipulation condition is:(5)$$H=\frac{1}{\hbar }{\mathrm{DS}}_{Z^{\prime }}^{2}+\frac{{g\mu }_{B}}{\hbar }B\cdot {S+H}_{\mathrm{e-phonon}}{+H}_{R}$$

It shows that the resonance coupling regulation of the NV center solid-state spins can be realized by phonon-coupled RF field and magnetic field.

### 2.2. Acoustic Characteristics Model of Phonon Piezoelectric Devices

During the phonon-coupled manipulation of the NV center, the acoustic wave propagates in the ZnO piezoelectric membrane layer, and its acoustic propagation characteristic is [36]:(6)$$\rho \frac{\partial^{2}u_{i}}{\partial t^{2}}{=c}_{\mathrm{ijkl}}\frac{\partial^{2}u_{k}}{\partial x_{l}\partial x_{j}}\;(i,\;j,\;k,\;l=1,2,3)$$
where ρ is the density of the ZnO piezoelectric membrane, $u_{i}$ is the displacement of the ZnO piezoelectric membrane along the direction of $x_{i}$, t is the time, and $c_{\mathrm{ijkl}}$ is the elastic stiffness matrix of the ZnO piezoelectric membrane.
(7)$$u_{i}{=\alpha }_{i}\exp\left\{j\left[\omega t-\beta \left(l_{1}x_{1}{+l}_{2}x_{2}{+l}_{3}x_{3}\right)\right]\right\}\;\left(i=1,2,3\right)\;$$
where $l_{1}$, $l_{2}$, and $l_{3}$ are the direction cosine of the wave propagation, ω=2πf is the angular frequency, and $\beta =\frac{2\pi }{\lambda }$; $\alpha_{i}$ is the amplitude of the sound wave. It can be obtained from Equations (6) and (7) that:(8)$$\left(c_{\mathrm{ijkl}}l_{l}l_{j}-{\rho v}^{2}\delta_{\mathrm{ik}}\right)\alpha_{k}=0$$
where $v^{2}=\frac{\omega^{2}}{\beta^{2}}$, $\delta_{\mathrm{ik}}=1(i=k)$ and $\delta_{\mathrm{ik}}=1(i\neq k)$. The elastic stiffness coefficient relationship in the ZnO piezoelectric membrane for the acoustic wave with a given propagation direction of the phonon piezoelectric device of the NV center is [26,36]:(9)$$c_{11}{=c}_{22}{=c}_{33};\;c_{12}{=c}_{13}{=c}_{21}{=c}_{23}{=c}_{31}{=c}_{32};\;c_{44}{=c}_{55}{=c}_{66}=1/2\left(c_{11}{-\;c}_{22}\right);\;c_{14}{=c}_{15}{=c}_{16}{=c}_{24}{=c}_{25}{=c}_{26}{=c}_{34}{=c}_{35}{=c}_{36}{=c}_{41}{=c}_{42}{=c}_{43}{=c}_{44}{=c}_{45}{=c}_{46}\;{=c}_{51}{=c}_{52}{=c}_{53}{=c}_{54}{=c}_{56}{=c}_{61}{=c}_{62}{=c}_{63}{=c}_{64}{=c}_{65}=0$$
because of the phonon propagating along the direction of $x_{i}$. $l_{1}=l_{2}=0$, $l_{3}$ can be solved as follows:(10)$$l_{3}^{\left(1\right)}{=l}_{3}^{\left(2\right)}=-j{\left(1-\frac{v^{2}}{v_{t}^{2}}\right)}^{1/2}\;\;l_{3}^{\left(3\right)}=-j{\left(1-\frac{v^{2}}{v_{l}^{2}}\right)}^{1/2}\;l_{3}^{\left(4\right)}{=l}_{3}^{\left(5\right)}=-j{\left(1-\frac{v^{2}}{v_{t}^{2}}\right)}^{1/2}\;\;l_{3}^{\left(6\right)}=-j{\left(1-\frac{v^{2}}{v_{l}^{2}}\right)}^{1/2}$$
where $v_{t}=\sqrt{c_{44}/\rho }$ is the shear wave propagation velocity in ZnO piezoelectric membrane layer, $v_{l}=\sqrt{c_{11}/\rho }$ is the longitudinal wave propagation velocity, and v is the propagation velocity of the acoustic wave. Assuming that the displacement of the acoustic wave–particle at an infinite distance from the surface of the ZnO piezoelectric membrane is zero, the wave velocity equation satisfies [17]:(11)$${\left[2-\frac{v^{2}}{v_{t}^{2}}\right]}^{2}=4{\left[2-\frac{v^{2}}{v_{t}^{2}}\right]}^{1/2}{\left[2-\frac{v^{2}}{v_{l}^{2}}\right]}^{1/2}$$

The particle displacement of the acoustic wave is as follows:(12)$$u_{1}=C\left[e^{{j\beta l}_{3}^{\left(3\right)}x_{3}}{-\mathrm{Ae}}^{{+j\beta l}_{3}^{\left(2\right)}x_{3}}\right]\exp\left[j\beta \left(x_{3}-\mathrm{vt}\right)\right]\;u_{2}=0\;u_{3}=-\mathrm{jC}(1-\frac{v^{2}}{v_{l}^{2}}{)}^{1/2}\;\left[e^{{j\beta l}_{3}^{\left(3\right)}x_{3}}-\frac{1}{A}e^{{j\beta l}_{3}^{\left(2\right)}x_{3}}\right]\;exp\left[j\beta \left(x_{1}-\mathrm{vt}\right)\right]$$
where $A={\left(1-\frac{v^{2}}{v_{l}^{2}}\right)}^{1/2}$. The phonon propagation characteristics of the phonon piezoelectric device, including the phase velocity *v* and the electromechanical coupling coefficient K^2^, are introduced as follows:

The phase velocity *v*. The phase velocity *v* of the phonon piezoelectric device of the NV center is related to the center frequency and the delay time. The symmetrical frequency ($f_{M-}$) and antisymmetrical frequency ($f_{M+}$) of the phonon piezoelectric device correspond to the upper and the lower boundary frequencies, respectively. The solution method for the phase velocity *v* of the phonon piezoelectric device is as follows:(13)$${v=f}_{0}\lambda \;f_{0}=\;(f_{\mathrm{sc}-}{+f}_{\mathrm{sc}-})/2=\;(f_{M-}{+f}_{M+})/2\;{v=(f}_{M-}{+f}_{M+})p$$

Electromechanical coupling coefficient K^2^. The electromechanical coupling coefficient of the phonon piezoelectric device is defined as the ratio of acoustic wave mechanical energy to excitation electrical energy, which can measure the acoustoelectric conversion efficiency. The electromechanical coupling coefficient K^2^ can be solved according to the resonant frequency and antiresonance frequency in the admittance characteristic:(14)$$K2=\left(f_{\mathrm{ar}}^{\;2}-f_{r}^{\;2}\right){/f}_{r}^{\;2}$$

### 2.3. Mathematical Construction of the Phonon Piezoelectric Device of the NV Center

(1)Model construction of the IDT of the phonon piezoelectric device

The IDT structure model of the phonon piezoelectric device of the NV center is shown in Figure 2. The characteristic function of the phonon field according to the Bragg reflection equation is as follows [36]:(15)β=π/p+q
(16)$$\{\begin{array}{c} \varphi_{+}{(x)=R(x)e}^{-j\pi x/p}\\ \varphi_{-}{(x)=S(x)e}^{j\pi x/p} \end{array}$$
where *β* is the acoustic wave number, *q* is the difference between the wave number and the crystal wave number, $\varphi_{+}\left(x\right)$ is the acoustic wave field in the +*x* direction, $\varphi_{-}\left(x\right)$ is the acoustic wave field in the −*x* direction, R(x) and S(x) are gradient field in space x, which satisfies the conditions as follows [36]:(17)$$\{\begin{array}{c} {R(x)=R}_{0}e^{-\mathrm{jqx}}\\ {S(x)=S}_{0}e^{-\mathrm{jqx}} \end{array}$$

Define the tuning parameters δ as:(18)δ=ω/ν−π/p−jγ
where ν is the acoustic wave velocity; γ is the acoustic wave loss. The differential decomposition equation of the phonon piezoelectric device is:(19)$$\{\begin{array}{c} \frac{\mathrm{dR}(x)}{\mathrm{dx}}=-j\delta R(x)+j\kappa S(x)+j\alpha V\\ \frac{\mathrm{dS}(x)}{\mathrm{dx}}{=-j\kappa }^{*}{R(x)+j\delta S(x)-j\alpha }^{*}V\\ \frac{\mathrm{dS}(x)}{\mathrm{dx}}{=-j2\alpha }^{*}R(x)-j2\alpha S(x)+\mathrm{jC}\omega V \end{array}$$
where $\kappa^{*}$ is the reflection coefficient of the phonon device, α is the transduction coefficient of the interdigital transducer, C is the capacitance, and V is the excitation amplitude voltage.

(2)Structural unit analysis of the phonon piezoelectric device

In this section, the finite element method is used to analyze the structural unit of the phonon piezoelectric device. The structural unit model of the phonon piezoelectric device of the NV center is shown in Figure 3a, which consists of an IDT unit layer, a ZnO piezoelectric membrane unit layer, and a diamond substrate unit. Structural unit parameters are set as follows: the IDT period λ is set as 0.5 μm; the IDT electrode width m is set as 0.25 μm; the IDT electrode thickness is set as 0.02 μm; the IDT interdigital width n is set as 0.25 μm; the diamond thickness h_1_ is set as 0.4 μm; the ZnO piezoelectric membrane thickness h_2_ is set as 0.4 μm. The piezoelectric membrane unit layer’s crystal structure of ZnO is shown in Figure 3b. The crystal orientation can be changed by setting the Euler angles (*α*, *β*, *γ*) of the ZnO piezoelectric membrane layer by rotating the coordinate system (as shown in Figure 3c).

The potential, Rayleigh, Sezawa, and Love wave distribution of the phonon piezoelectric device are shown in Figure 4a–d. The potential distribution on the surface of the ZnO piezoelectric membrane layer is uneven. The displacement components of the Rayleigh wave, Sezawa wave, and Love wave are concentrated in the *x*-direction, *z*-direction, and *y*-direction. Under the incentive conditions in this paper, the mechanical displacement field of the structural unit of the phonon piezoelectric device is in the magnitude of 10^−10^ m. The ZnO piezoelectric membranes with Euler angles of (0°, 0°, 0°) and (0°, 90°, 0°) were selected, respectively. The input conductance characteristics and resonance mode shapes of the structural unit of the phonon piezoelectric device are studied, as shown in Figure 5. It can be seen from Figure 5a that the first mode and fourth mode of the phonon piezoelectric device with Euler angles of (0°, 0°, 0°) are the Rayleigh wave. Their resonance frequencies are 952 MHz and 1248 MHz. The second mode and the third mode are the Sezawa mode. Their resonance frequencies are 1039 MHz and 1130 MHz. In Figure 5b, the acoustic wave propagation direction of the phonon piezoelectric device with Euler angles of (0°, 90°, 0°) is perpendicular to the *z*-axis. The first two resonant vibration modes are Love modes, and their resonant frequencies are 485 MHz and 1464 MHz. The acoustic wave fluctuation in the phonon piezoelectric device is much larger than in the diamond. Different Euler angles will affect the excitation and propagation characteristics.

## 3. Structural Optimization of Phonon Piezoelectric Device of the NV Center

### 3.1. Finite Element Model of the Phonon Piezoelectric Device

(1)Mathematical model

In this section, the finite element analysis method is used to simulate the propagation characteristic of the phonon piezoelectric device of the NV center. It is solved by the element matrix, which can be established using the type function, piezoelectric equation, variational equation, and virtual work equation at the element nodes. The mechanical characteristic of the phonon piezoelectric device of the NV center is expressed as:(20)$$M{u(t)\ddot{|}}_{{t=t}_{1}}{+D}_{\mathrm{uu}}{u(t)\dot{|}}_{{t=t}_{1}}{+K}_{\mathrm{uu}}{u(t)|}_{{t=t}_{1}}{=F}_{B}{+F}_{S}{+F}_{P}$$

The matrix equation of the piezoelectric coupling field of the phonon piezoelectric device of the NV center is:(21)$$K_{u\varphi }^{T}{u(t)|}_{{t=t}_{1}}{+K}_{u\varphi }{\varphi (t)|}_{{t=t}_{1}}{=Q}_{S}{+Q}_{P}$$

In Equations (20) and (21), *M* is the mass matrix of the ZnO phonon piezoelectric membrane, $K_{\mathrm{uu}}$ is the mechanical body force matrix, $F_{B}$ is the body force matrix, $F_{S}$ is the mechanical surface force matrix, $F_{P}$ is the mechanical point force matrix, $D_{\mathrm{uu}}$ is the mechanical damping matrix, $K_{u\varphi }$ is the piezoelectric coupling matrix, $Q_{S}$ is the surface charge matrix, $Q_{P}$ is the point charge matrix, u is the displacement of the ZnO piezoelectric membrane, φ is the electric potential, ${u(t)|}_{{t=t}_{1}}$ is the nodal displacement in the natural mode for $t_{1}$ time, and ${\varphi (t)|}_{{t=t}_{1}}$ is the electric potential for $t_{1}$ time.

Regardless of the mechanical damping, let $D_{\mathrm{uu}}=0$, and φ is eliminated. The matrix equation of the ZnO piezoelectric membrane of the phonon piezoelectric device is:(22)$$M{u(t)\ddot{|}}_{{t=t}_{1}}{+(K}_{\mathrm{uu}}{-K}_{u\varphi }K_{u\varphi }^{T}{)u(t)|}_{{t=t}_{1}}{=F}_{B}{+F}_{S}{+F}_{P}{-K}_{u\varphi }\left(Q_{S}{+Q}_{P}\right)$$

The homogeneous solution of Equation (22) corresponds to the various modes in the model and the frequencies corresponding to the various modes. The standard solution of u(t) is:(23)$$u(t)=u\cdot e^{j\omega t}$$
where *u* is the nodal displacement matrix in natural mode, and ω is the modal frequency. According to Equations (22) and (23), it can obtained that:(24)$$u\left({-\omega }^{2}{M+K}_{\mathrm{uu}}{-K}_{u\varphi }K_{u\varphi }^{T}\right)\cdot e^{j\omega t}=0$$

Its standard solution satisfies:(25)$$\left|{-\omega }^{2}{M+K}_{\mathrm{uu}}{-K}_{u\varphi }K_{u\varphi }^{T}\right|=0$$

The modal frequency ω can be solved by Equation (25). Its modal displacement matrix u can be solved by Equation (24).

(2)Finite element model and boundary conditions

The acoustic field of the phonon piezoelectric device of the NV center was simulated at the frequency range of [600,1600] MHz. The material of the IDT is copper. The substrate material of the piezoelectric membrane layer is the ZnO crystal. The structural unit type is set as a hexahedral eight-node element. The geometric grid model and the mechanical and electrical boundary conditions of the phonon piezoelectric device of the NV center are shown in Figure 6. The mesh size of the phonon piezoelectric device is set to 0.5 μm. The electrode part is divided by the mapping grid. Five grid cells are scanned along the thickness direction of the IDT. Face 1 is the voltage excitation boundary condition. Face 2 is the reference charge boundary condition. Face 3 is the reference voltage boundary condition. Face 4 is the fixed boundary condition. Face 5 is the perfect boundary layer. Simulation structure parameters of the phonon piezoelectric device of the NV center are shown in Table 1.

(3)Simulation results of phonon field characteristics

The phonon field properties of the phonon piezoelectric device were analyzed on the basis of the modal method. Its modal frequency is set as [600,1600] MHz, and its step size is set as 100 MHz. Secondly, the modal superposition method is used to analyze the harmonic response of the geometric model with different IDT spacings. The modal analysis and the harmonic response analysis results of the phonon device geometric model are shown in Figure 7 and Figure 8.

Figure 7 shows that the displacement inside the diamond substrate is negligible relative to the mechanical displacement of ZnO. The acoustic wave energy decays rapidly in the thickness direction of the substrate. It is concentrated on two wavelength scales. The acoustic wavelengths in the ZnO piezoelectric layer are 5.3 × 10^−6^ m, 4.2 × 10^−6^ m, 3.2 × 10^−6^ m, 2.7 × 10^−6^ m, 2.4 × 10^−6^ m, and 2.2 × 10^−6^ m at the resonant frequencies of 700 MHz, 800 MHz, 900 MHz, 1000 MHz, 1100 MHz, and 1200 MHz. The maximum displacements of the acoustic waves are 0.1 nm, 0.3 nm, 0.7 nm, 0.8 nm, 2.5 nm, and 1.4 nm, respectively. Figure 8a–d correspond to the average particle displacements at the interface of the diamond and ZnO piezoelectric membrane inside the diamond at the IDT excitation tail and the IDT receiving tail at the IDT space of 1 μm. Simultaneously, the ZnO piezoelectric layer of 2 μm and 3 μm IDT space are also simulated. The resonant frequency of the phonon piezoelectric device is 1096 MHz at the IDT space of 1 μm. The maximum displacement at the interface of the diamond and ZnO is 5.19 × 10^−10^ m. The maximum displacement of the IDT receiving tail is 1.58 × 10^−8^ m, which is much larger than that inside the diamond. The phonon piezoelectric device with the IDT space of 1 μm, 2 μm, and 3 μm has resonant frequencies of 1096 MHz, 657 MHz, and 412 MHz, respectively. According to the literature opened by our team before [21], the phonon state density e for different IDT space is: $e_{1\mu m}{:\;e}_{2\mu m}{:e}_{3\;\mu m}\approx 1:0.5:0.3$.

### 3.2. Structural Optimization of the Phonon Piezoelectric Device

(1)Geometry parameters

IDT electrode positions are optimized on the basis of the structural model of the phonon piezoelectric device in Section 3.1. Phonon piezoelectric devices with different electrode positions, including four cases (shown in Figure 9), were analyzed. In addition, the (100) ZnO with Euler angles of (0°, 0°, 0°) and the (002) ZnO with Euler angles of (0°, 90°, 0°), respectively, were analyzed. The specific description is as follows: Case 1: the IDT electrode is located on the ZnO piezoelectric membrane layer. Case 2: the IDT electrode is located between the diamond and the ZnO piezoelectric membrane layer. Case 3: the ZnO piezoelectric membrane material is (100) ZnO. Case 4: the ZnO piezoelectric membrane material is (002) ZnO. The IDT electrode (thickness of 0.1 μm) is located on the surface of the ZnO piezoelectric membrane in both Case 3 and Case 4. The material parameters of the phonon piezoelectric device of the NV center are shown in Table 2. The boundary conditions are the same as those in Section 3.1. The optimized parameters of the IDT electrode size used are shown in Table 3.

(2)Optimization results

The acoustic characteristics of the phonon piezoelectric device are listed in Section 3.1. The phase velocity *v* and the electromechanical coupling coefficient K^2^ are analyzed in this section. The electrode input admittance characteristics of the phonon piezoelectric device of the NV center for different cases are analyzed, as shown in Figure 10. Four peaks can be observed, the first peak is the low-order mode (M11 mode) Rayleigh wave, the second peak is the low-order mode (M21 mode) Sezawa wave, and the third peak is the high-order mode (M22 mode) Sezawa wave, and the fourth peak is the high-order mode (M12 mode) Rayleigh wave. The modal waveforms for different cases are shown in Figure 11.

The relationship between the phase velocity characteristics and the thickness of the IDT layer for different cases is shown in Figure 12a,b. The phase velocities of the M11 mode and M12 mode of the phonon piezoelectric device are greater than those of the M21 mode and M22 mode. The phase velocity of the Sezawa wave mode is smaller when the IDT layer is located between the ZnO piezoelectric membrane and the diamond. The phase velocity characteristic curves for different thicknesses of the ZnO piezoelectric membrane layer and the Euler angle are shown in Figure 12c,d. In Case 3 and Case 4, the phase velocity of the phonon piezoelectric device decreases with the increase in the thickness of the ZnO piezoelectric membrane layer. The phase velocities in M11 mode and M12 mode are larger than those in M21 mode and M22 mode.

The potential oscillation period of the phonon piezoelectric device for Case 1 and Case 2 are shown in Figure 13. The potential of the structural unit of the phonon piezoelectric device changes periodically with the excitation frequency. The potential oscillation period is divided into eight stages, and a high-potential region and a low-potential region appear in Case 1. The potential in the first four stages gradually evolves from single mode to multimode, and the potential in the latter four stages gradually evolves from multimode to single mode. In Case 2, the potential oscillation period is divided into twelve stages, and the evolution periods are single mode, multimode, one-time inversion, single mode, multimode, and second-time inversion.

The electric field mode and current density mode of the phonon piezoelectric device for Case 1 and Case 2 are shown in Figure 14. The electric field mode and the current density mode are shear wave modes of oscillatory variation along the *x*-direction for Case 1. The electric field mode inside the ZnO piezoelectric membrane layer is a single-mode state, and the current density mode is a longitudinal wave mode of oscillation along the *x*-direction for Case 2. The electromechanical coupling coefficient K^2^ for different cases is shown in Figure 15. The electromechanical coupling coefficient K^2^ of the M12 mode, M22 mode, M21 mode, and M11 mode for Case 2 is larger than those for Case 1. The electromechanical coupling coefficient K^2^ of the phonon piezoelectric device in the Sezawa wave coupled mode increases and then decreases with the increase in the thickness of the ZnO piezoelectric membrane layer for Case 3 and Case 4. The electromechanical coupling coefficient K^2^ of (100) ZnO phonon device is larger than that of (002) ZnO phonon device. When the thickness of the (100) ZnO piezoelectric membrane layer is [400,600] nm, it has optimized electromechanical coupling coefficients for four modes.

## 4. Phonon-Coupled Manipulation Experiment of the NV Center

### 4.1. MEMS Implement of the Phonon Piezoelectric Device of the NV Center

The phonon piezoelectric device of the NV center is designed on the basis of the optimized structure in Section 3. The size of the diamond with the NV center is 4800 μm × 4400 μm × 1000 μm. The size of the (100) ZnO is 4600 μm × 4400 μm × 400 μm. The width of the interdigital is 3 μm. The width of the interdigital gap is 6 μm. The thickness of the interdigital gap is 300 nm. The thickness of the electrode is 1 μm. The MEMS structure size of the phonon piezoelectric device of the NV center is shown in Figure 16.

We use MEMS technology to prepare the phonon piezoelectric device of the NV center. Figure 17a shows the specific MEMS implementation process: (1) Clean the diamond samples; (2) ZnO surface coated with photoresist; (3) photoresist surface covered with interdigitated electrode pattern; (4) exposure and development; (5) deposit the copper membrane, and the thickness ratio of the titanium adhesion layer and the copper is 1:5; (6) lift off—the sample was placed in an acetone solution, and the photoresist and the metal layer above are stripped off under the ultrasonic waves; (7) Bonding of ZnO to diamond. The prepared phonon piezoelectric device of the NV center was characterized by an M330-HK830 optical micrograph (Its manufacturer is AOSVI, which is located in Shenzhen, China), as shown in Figure 17b.

### 4.2. Phonon-Coupled Manipulation Experiment

(1)Experimental environment and platform

The temperature, humidity, and air pressure of the NV center measurement carried out are in natural conditions. The experimental platform is the quantum diamond single-spin spectrometer system (Its manufacturer is CIQTEK, which is located in Hefei, China), as shown in Figure 18. This system is divided into multiple submodules, including the optical module, probe module, magnet module, microwave module, arbitrary sequence generation module, radio frequency module, main control module, power module, optical stabilization platform, workstation, air compressor, and other parts. The optical platform is used for the vibration isolation of the optical path and the probe module. Microwave modules, power modules, main control modules, and arbitrary sequence generation modules are placed in the control cabinet. The diamond sample, the radiation module, the probe module, and the magnet module interact with the control cabinet through the computer. The block diagram of the quantum diamond single-spin spectrometer system is shown in Figure 19.

The prepared phonon piezoelectric device is coupled to the probe module. It is excited by a radio frequency signal generator through a phonon excitation circuit. The schematic diagram of the architecture of the phonon-coupled manipulation module is shown in Figure 20.

The schematic diagram and physical photograph of the phonon-coupled radiation structure are shown in Figure 21a,b. The radio frequency signal generator generates a phonon excitation source by amplifying the radio frequency signal. The phonon-coupled radiation structure is connected to the radio frequency amplifier through the phonon excitation conduction plate and the phonon excitation input SMA port. The phonon device of the NV center is placed on the glass slide above the optical hole of the radiation structure and is the core component of the phonon-coupled manipulation module.

(2)Experimental method

The experimental method includes six steps: (1) Characterizing the diamond samples. The diamond samples were optically characterized and selected with fewer impurities and higher purity (more than 99%). (2) Coarse adjusting the position of the NV center. Coarse adjustment is accomplished by adjusting the position of the micrometer stage to focus the laser on the surface of the diamond. (3) Selecting an appropriate NV center. The laser irradiated the diamond continuously. We focused the fluorescence position on the NV center by nanostage fine-tuning the position of the diamond. (4) Optical detection magnetic resonance (ODMR) measurement (the sequence is shown in Figure 22a). (5) Rabi measurement. Firstly, the NV center was radiated by a microsecond continuous laser pulse. Secondly, the spin state was polarized to $m_{s}=0$. A microsecond was waited to let the electronic return to the ground state. Finally, the resonance microwave of τ duration is loaded, and the laser pulse is used to perform self-rotational reading (the sequence is shown in Figure 22b). (6) Ramsey measurement. A few microseconds of continuous laser pulse were applied to the diamond. Then, a pause for 1 μs to initialize the spin state to the ground state. The π/2 pulsed microwave (Measured by the Rabi oscillation experiment) was loaded to transform the spin state into a coherent superposition state. After the free evolution of time τ, the relative phase of superposition accumulation ϕ=2πγB/τ. The second pulsed microwave was loaded to project the accumulated phase information of spin states “|0〉” and “|1〉”, thereby inferring the relative population of spins. The ODMR spectrum is obtained by continuously changing the microwave frequency and collecting fluorescence counts. Two fluorescence counts have been collected: the signal value (SIG) and the reference value (REF). The signal value is the fluorescence count corresponding to the quantum state readout after microwave manipulation. The reference value is the fluorescence count corresponding to the direct readout of the fluorescence signal after the laser initializes the quantum state. Contrast is defined as (SIG-REF)/REF to reduce the error.

(3)Experimental results

Characterization of the diamond with the NV center is shown in Figure 23. It has superior manipulation and measurement capabilities when selecting a single NV center point in a relatively pure area near the copper wire. The fluorescence in the range centered on the single NV center is scanned and counted along with the X, Y, and *Z*-axis directions, and the fluorescence scanning curve is also shown in Figure 23. It has a distinct unimodal characteristic, which indicates that the selected NV center is effective.

We applied 350 MHz acoustic excitation and a 4.6 G magnetic field along the NV axis. Time-domain measurement at the MW frequency of 2870 MHz and the ODMR spectrum for different cases are shown in Figure 24, including the non-phonon-coupled case and the phonon-coupled case (including Case 1 and Case 2). In Case 1, IDT is deposited on the surface of the (100) ZnO piezoelectric membrane layer. In Case 2, IDT is deposited on the interface of the diamond and (100) ZnO piezoelectric membrane. The time-domain measurement result shows that the phonon-coupled case with the average fluorescence count at the MW frequency of 2870 MHz is about 992 within 100 s. That count for the non-phonon-coupled case in the same condition is about 982. It can be shown from the ODMR spectrum that the $m_{s}=\pm 1$ state is split into $m_{s}=-1$ and $m_{s}=+1$. The microwave resonance frequencies for the non-phonon-coupled case are 2857.62 MHz and 2882.44 MHz. The microwave resonance frequencies for the phonon-coupled case are about 2857 MHz and 2882 MHz, both of which are symmetrical, about 2870 MHz. However, the fluorescence intensity for the phonon-coupled case is higher than that of the non-phonon-coupled case. In Case 1 and Case 2, the average fluorescence intensity increased by 1.02% and 1.29%, respectively, compared with the non-phonon-coupled case. The possible reason is that more photons jump with phonon-coupled manipulation. This phenomenon is more obvious around microwave frequencies. The bandwidth of the resonance peak for the phonon-coupled case is larger. Compared with Case 2, the fluorescence intensity is more unstable for Case 1.

The Rabi measurement results for the phonon-coupled case and the non-phonon-coupled case are shown in Figure 25. The π pulse time of the single NV center for the phonon-coupled case is 148.04256 ns, and that for the non-phonon-coupled case is 148.998 ns. The Rabi oscillation amplitude increased by 1.14% for the phonon-coupled case. Distortion microwave frequency 2867 MHz was applied to measure the Ramsey oscillation for the phonon-coupled case and non-phonon-coupled case (see Figure 11). In these two cases, Ramsey oscillation rendered a good cycle characteristic, as shown in the fitting function in Figure 26. Compared with the non-phonon-coupled case, the Ramsey oscillation for the phonon-coupled case has a greater oscillation amplitude (about 2.36% higher), bringing an obvious phase recognition effect. The phase shift effect is not obvious. In addition, it can be seen from the attenuation coefficient that the decoherence time for the phonon-coupled case is higher than the non-phonon-coupled case, improving the validity of spin measurement.

According to Figure 24, Figure 25 and Figure 26, the ODMR linewidth $\Gamma_{\mathrm{ODMR}}$ and contrast $C_{\mathrm{ODMR}}$ are expressed as follows:(26)$$\Gamma_{\mathrm{ODMR}}{=\Gamma }_{\mathrm{ODMR}}^{\left(\mathrm{inh}\right)}+\Gamma_{\mathrm{ODMR}}^{\left(h\right)}\left(1+\frac{{4\pi \kappa P}_{\mathrm{MW}}}{\Gamma_{\mathrm{ODMR}}^{\left(h\right)}\gamma_{1}}\right)$$
(27)$$C_{\mathrm{ODMR}}=C_{\mathrm{ODMR}}^{\left(\max\right)}\frac{{4\pi \kappa P}_{\mathrm{MW}}}{{4\pi \kappa P}_{\mathrm{MW}}{+\Gamma }_{\mathrm{ODMR}}^{\left(h\right)}\gamma_{1}}$$
where $\Gamma_{\mathrm{ODMR}}^{\left(h\right)}=7.8\;\mathrm{MHz}$ and $\Gamma_{\mathrm{ODMR}}^{\left(\mathrm{inh}\right)}\;=0.5\;\mathrm{MHz}$ are the homogeneous and inhomogeneous line widths in the absence of power broadening. $P_{\mathrm{MW}}=-6\;\mathrm{dBm}$ (0.251 mW) is the MW power, κ ≈ 20 MHz is a proportionality factor such that ${\kappa P}_{\mathrm{MW}}$ is the spin Rabi frequency, and $\gamma_{1}\approx \;22\;\mathrm{MHz}$ is the effective spin relaxation rate. The population transfer rate I is obtained:(28)$$I=\frac{1}{3}{\left(\frac{\Omega_{0}{+\Omega }_{P}}{k_{B}T}\right)}^{3}{\left(1-\frac{h\xi }{\Omega_{0}{+\Omega }_{P}}\right)}^{2}\left(1+\frac{h\xi }{2\left(\Omega_{0}{+\Omega }_{P}\right)}\right)$$
where $k_{B}=1.38\;\times \;10^{-23}\;J/K$ is the Boltzmann constant; T=298.15 K is the room temperature; h = 6.63 ×$10^{-34};$ J·s is the Planck constant; ξ is the strain splitting; hξ=4.6 meV; $\Omega_{0}=\gamma B$ is the actual Larmor frequency, γ is the gyromagnetic ratio, B=80 G is the magnetic field along the NV axis; $\Omega_{P}{=g}_{2}\sqrt{\langle n\rangle }\cdot \Omega_{0}{/\omega }_{m}$ is effective Larmor frequency of the spin transition for phonon-coupled manipulation case; 〈n〉 is the average number of phonons; $\Omega_{0}{+\Omega }_{P}=\left({\;\omega }_{m}\Omega_{0}{+g}_{2}\sqrt{\langle n\rangle }\cdot \Omega_{0}\right){/\omega }_{m}$ is the effective Larmor frequency of the spin transition; $g_{2}{=\delta \varepsilon }_{c}k_{m}\sqrt{\hbar {/2\;m\omega }_{m}}$ is the electron-phonon effective coupling rate; $\varepsilon_{c}$ is the band edge energy during lattice vibration; ${\delta \varepsilon }_{c}=\left(\frac{\partial \varepsilon_{c}}{\partial V}\right)\delta V$ is the deformation potential; δV is the local variation of the longitudinal acoustic mode of the lattice vibration; $k_{m}$ is the wavenumber of the phonon mode; m is the effective mechanical mass oscillator; $\omega_{m}$ is the phonon oscillation frequency. Because the phonon state density e for different IDT space is: $e_{1\mu m}{:e}_{2\mu m}{:e}_{3\mu m}\approx 1:0.5:0.3$, number of phonons 〈n〉=ke. $\left(\Omega_{0}\right){\;}_{\mathrm{non-phonon}}:\;\left(\Omega_{0}{+\Omega }_{P}\right){\;}_{1\;\mu m}:\;\left(\Omega_{0}{+\Omega }_{P}\right){\;}_{2\;\mu m}:\;\left(\Omega_{0}{+\Omega }_{P}\right){\;}_{3\;\mu m}\approx 1:\;1.0102:\;1.0071:\;1.0055$. According to Equation (25), the population transfers rate for different IDT space is: ${I\;}_{\mathrm{non-phonon}}{:\;I\;}_{1\;\mu m}{:\;I\;}_{2\;\mu m}{:\;I\;}_{3\;\mu m}\approx 1:\;1.0309:\;1.0214:\;1.0166$. The fluorescence transition efficiency at the IDT space of 1 μm, 2 μm, and 3 μm for Case 2 increased by 3.09%, 2.14%, and 1.66%, respectively. It is basically consistent with the experimental results. So, the phonon resonance manipulation method can effectively increase the NV center’s spin transition probability with the MEMS phonon piezoelectric device prepared in this paper, improving the quantum spin manipulation efficiency.

It is also obtained that the fluorescence measurement error $\sigma_{P}$ for phonon-coupled manipulation case is:(29)$$\sigma_{P}=\frac{S_{m}}{{S+S}_{m}}\cdot \frac{{\;\Omega }_{0}}{\Omega_{0}{+\Omega }_{P}}$$
where S is the measurement value without environmental disturbance and $S_{m}$ is the extra measurement value brought by environmental disturbance, so the error caused by external magnetic field interference can be effectively compensated by the phonon-coupled control method.

## 5. Conclusions

The phonon piezoelectric device of the NV center is designed on the basis of the phonon-coupled regulation mechanism. The propagation characteristics and acoustic wave excitation modes of the phonon piezoelectric device are analyzed. In order to improve the performance of phonon-coupled manipulation, the influence of the structural parameters of the diamond substrate and the ZnO piezoelectric layer on the phonon propagation characteristics are analyzed. The structure of the phonon piezoelectric device of the NV center is optimized, and its MEMS implementation and characterization are carried out. Research results show that the electromechanical coupling coefficient K^2^ of (100) ZnO phonon device is larger than that of (002) ZnO phonon device. When the thickness of the (100) ZnO piezoelectric membrane layer is [400,600] nm, it has optimized electromechanical coupling coefficients for four modes. More photons jump with phonon-coupled manipulation. This phenomenon is more obvious around microwave frequencies. The bandwidth of the resonance peak for the phonon-coupled case is larger. The fluorescence intensity is stable for the case of IDT deposited on the surface of the (100) ZnO piezoelectric membrane layer. The fluorescence transition efficiency at the IDT space of 1 μm, 2 μm, and 3 μm for Case 2 increased by 3.09%, 2.14%, and 1.66%, respectively. The phonon resonance manipulation method can effectively increase the NV center’s spin transition probability with the MEMS phonon piezoelectric device prepared in this paper, improving the quantum spin manipulation efficiency.

## Figures and Tables

**Figure 1 micromachines-13-01628-f001:**
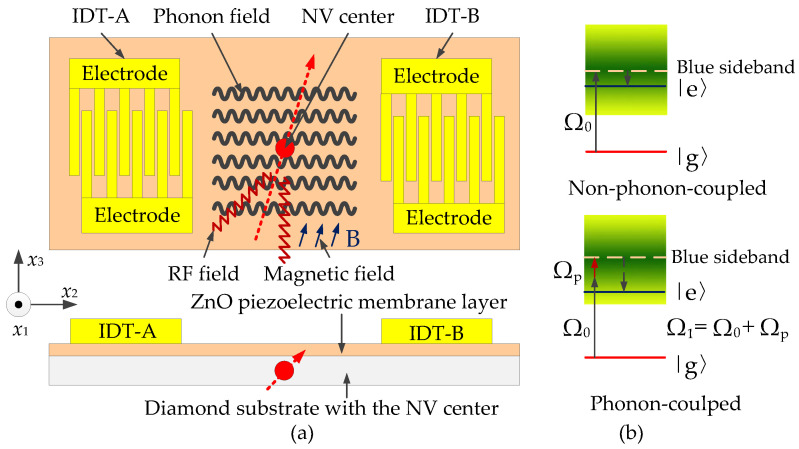
Phonon-coupled resonance structure and manipulation mechanism of the NV center. (**a**) Phonon-coupled resonance structure; (**b**) phonon-coupled manipulation mechanism.

**Figure 2 micromachines-13-01628-f002:**
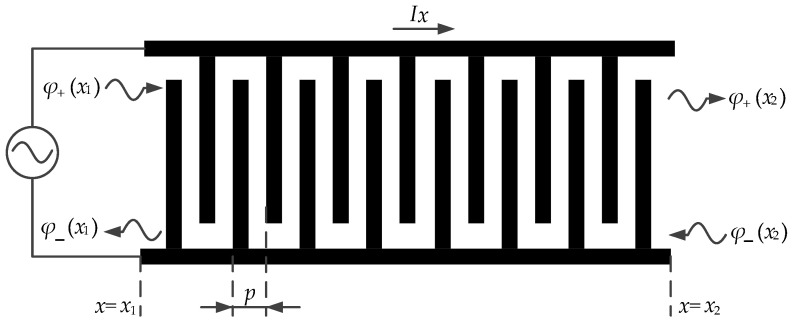
IDT structure model of the phonon piezoelectric device of the NV center.

**Figure 3 micromachines-13-01628-f003:**
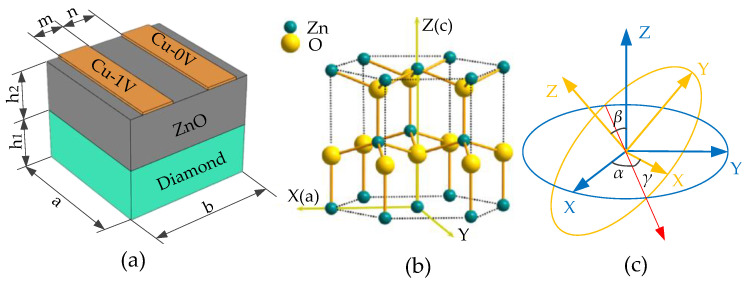
Structural unit model of phonon piezoelectric device and ZnO microstructure. (**a**) Structural unit model of phonon piezoelectric device; (**b**) the crystal structure of ZnO; (**c**) a plot of rotation coordinate of ZnO.

**Figure 4 micromachines-13-01628-f004:**
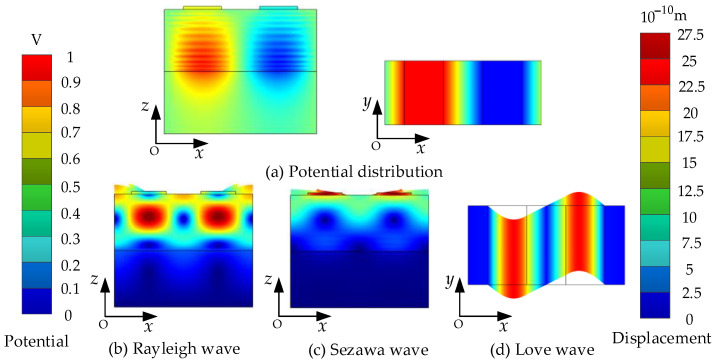
Potential, Rayleigh, Sezawa, and Love wave distribution of the phonon piezoelectric device. (**a**) Potential distribution; (**b**) Rayleigh wave distribution; (**c**) Sezawa wave distribution; (**d**) Love wave distribution.

**Figure 5 micromachines-13-01628-f005:**
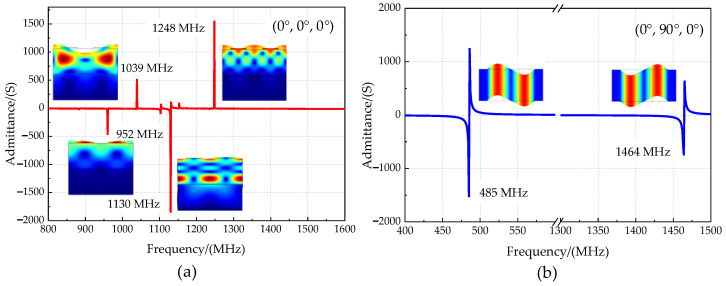
Admittance characteristics for different crystalline phonon device structural units. (**a**) (0°, 0°, 0°) ZnO; (**b**) (0°, 90°, 0°) ZnO.

**Figure 6 micromachines-13-01628-f006:**
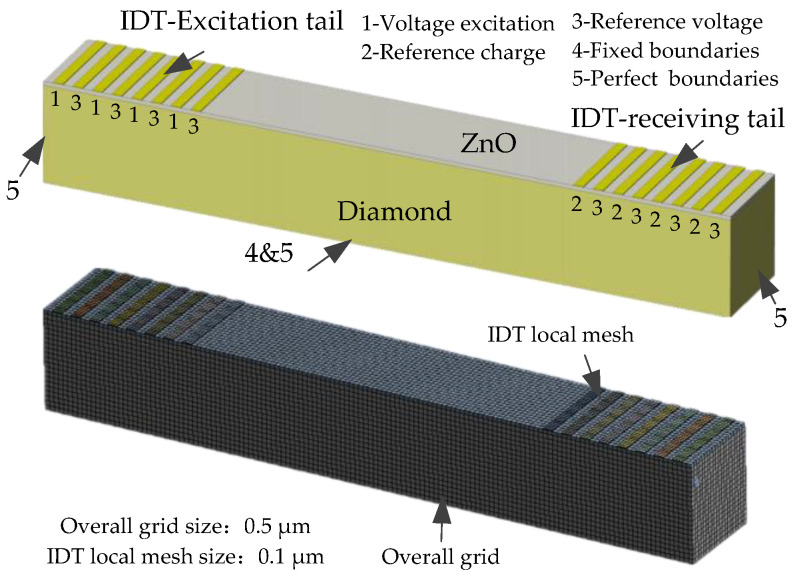
The geometric grid model and boundary conditions of the phonon piezoelectric device.

**Figure 7 micromachines-13-01628-f007:**
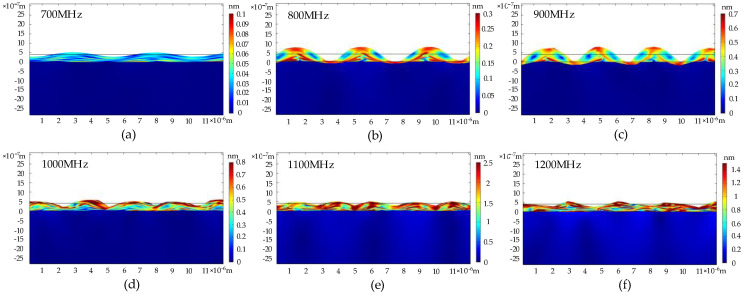
Phonograph displacement field in different resonant frequencies of the phonon piezoelectric device of the NV center. (**a**) 700 MHz; (**b**) 800 MHz; (**c**) 900 MHz; (**d**) 1000 MHz; (**e**) 1100 MHz; (**f**) 1200 MHz.

**Figure 8 micromachines-13-01628-f008:**
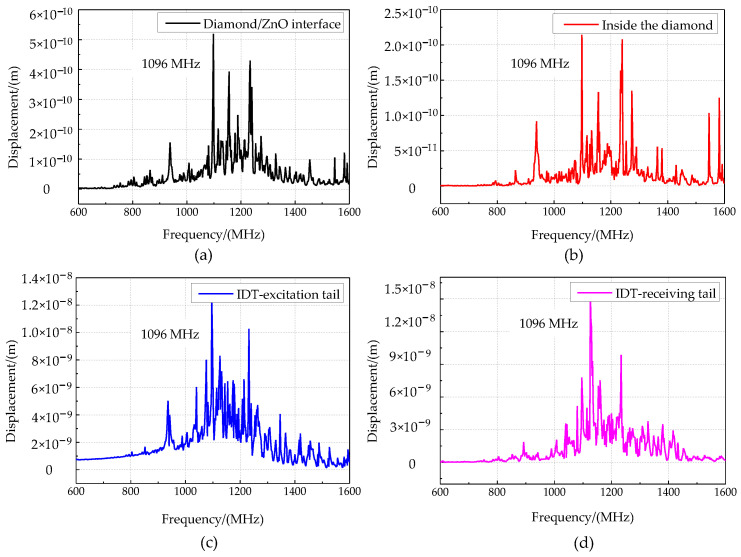
Relationship between displacement and frequency of the phonon piezoelectric device. (**a**) Diamond–ZnO interface; (**b**) inside the diamond; (**c**) IDT excitation tail; (**d**) IDT receiving tail.

**Figure 9 micromachines-13-01628-f009:**
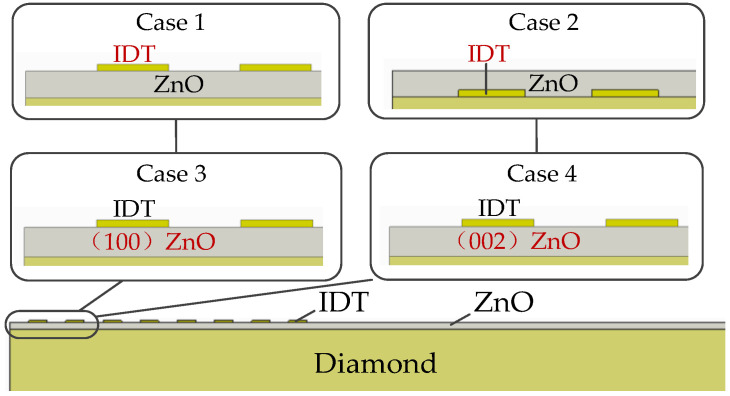
Phonon piezoelectric device structure of the NV center for 4 cases.

**Figure 10 micromachines-13-01628-f010:**
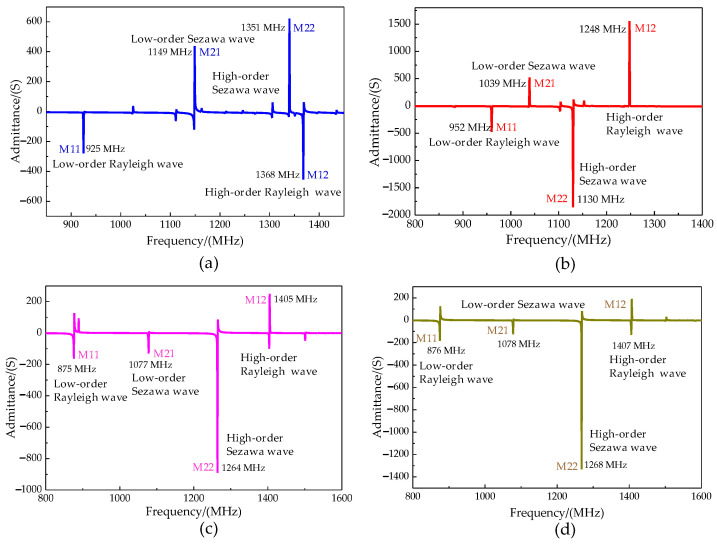
Electrode admittance characteristics of the phonon piezoelectric device of NV center for different cases. (**a**) Case 1a. (**b**) Case 1b. (**c**) Case 2a. (**d**) Case 2b.

**Figure 11 micromachines-13-01628-f011:**
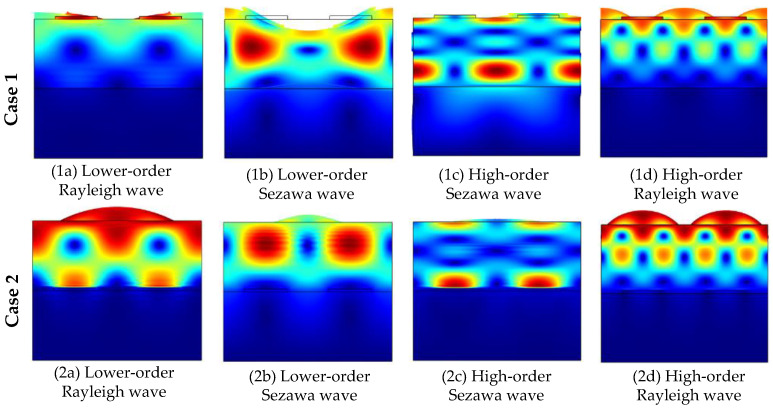
Modal waveforms for different cases.

**Figure 12 micromachines-13-01628-f012:**
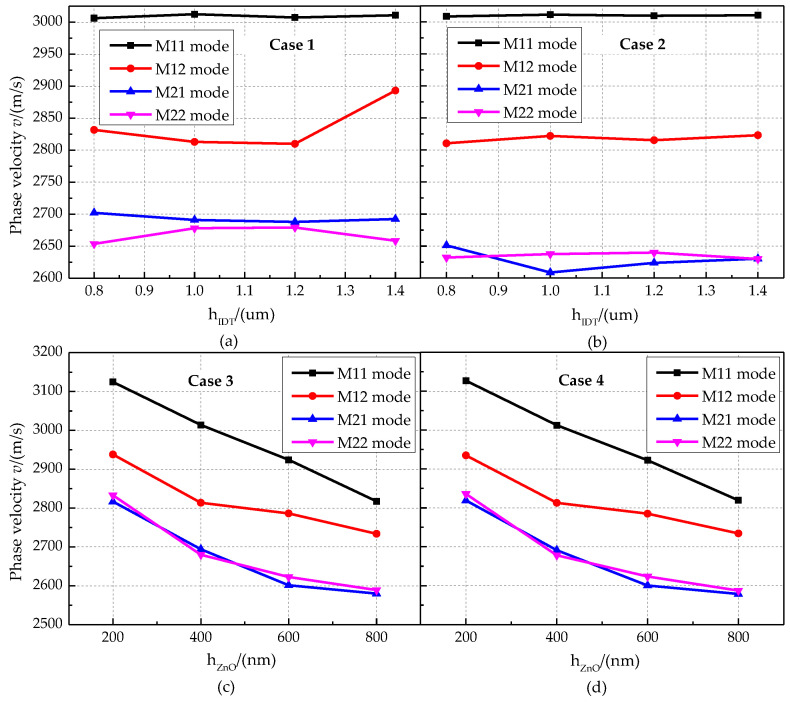
Phase velocity characteristics of the phonon piezoelectric device for different cases. (**a**) Case 1. (**b**) Case 2. (**c**) Case 3. (**d**) Case 4.

**Figure 13 micromachines-13-01628-f013:**
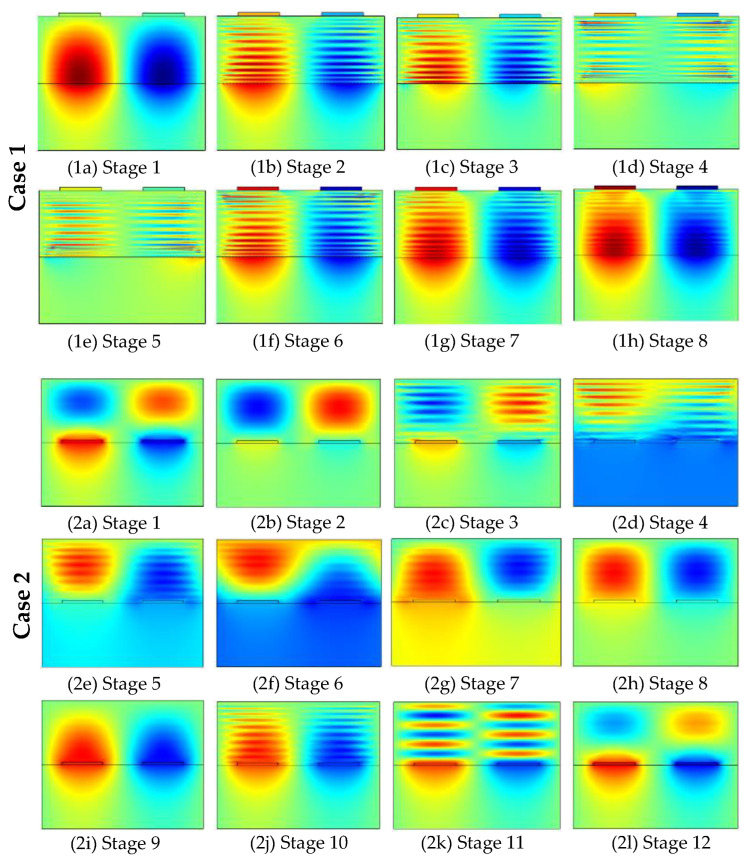
Potential oscillation period of the phonon piezoelectric device.

**Figure 14 micromachines-13-01628-f014:**
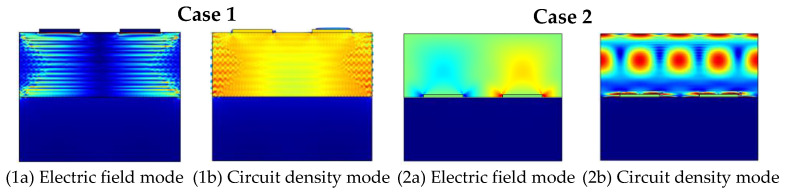
Electric field mode and current density mode of the phonon piezoelectric device.

**Figure 15 micromachines-13-01628-f015:**
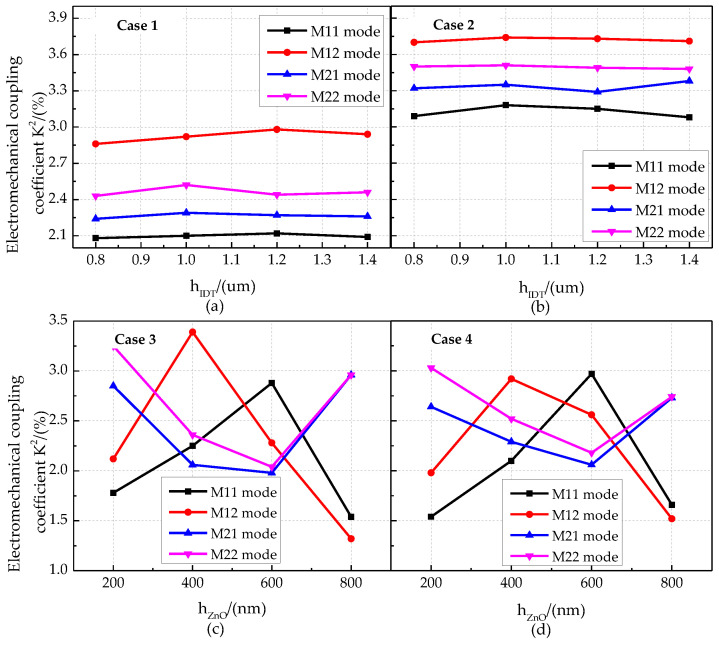
Electromechanical coupling coefficient K^2^ for different cases. (**a**) Case 1. (**b**) Case 2. (**c**) Case 3. (**d**) Case 4.

**Figure 16 micromachines-13-01628-f016:**
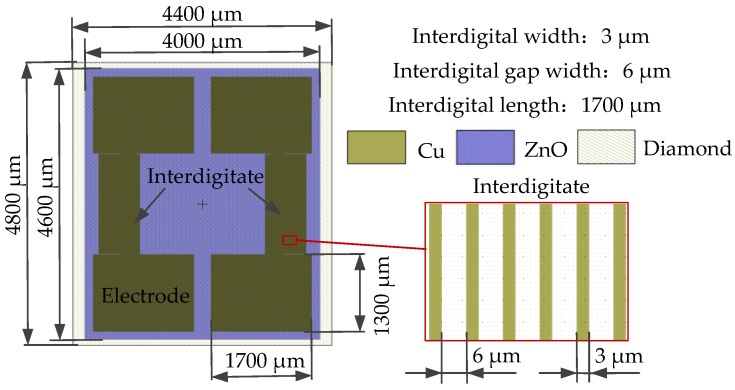
The MEMS structure size of the phonon piezoelectric device of the NV center.

**Figure 17 micromachines-13-01628-f017:**
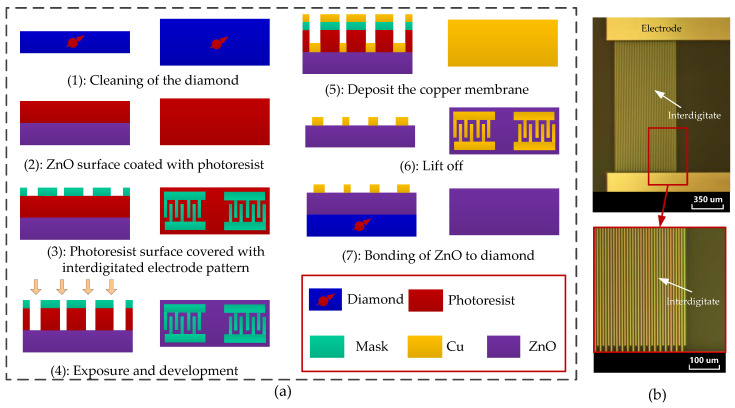
MEMS implementation of the phonon piezoelectric device of the NV center. (**a**) MEMS process. (**b**) Optical micrograph.

**Figure 18 micromachines-13-01628-f018:**
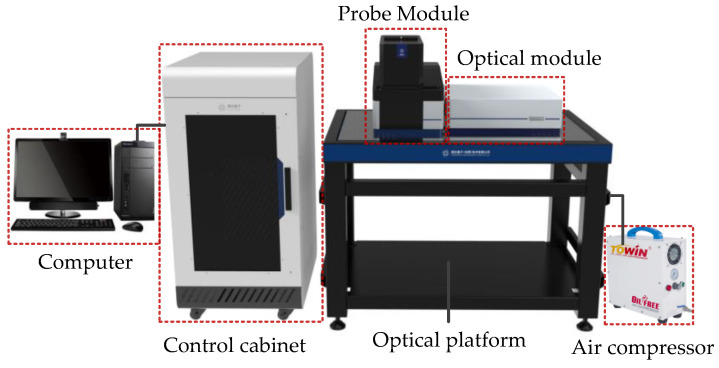
Quantum diamond single-spin spectrometer system.

**Figure 19 micromachines-13-01628-f019:**
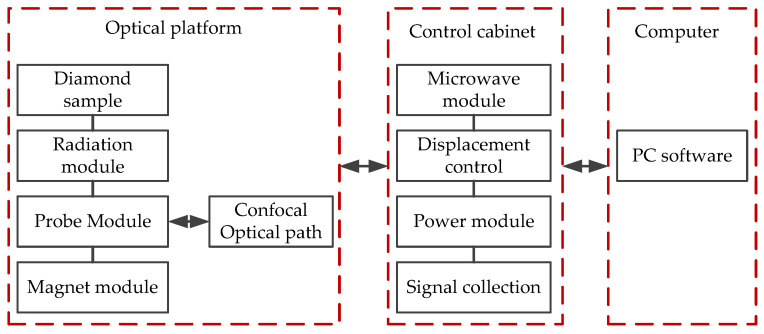
Block diagram of the quantum diamond single-spin spectrometer system.

**Figure 20 micromachines-13-01628-f020:**
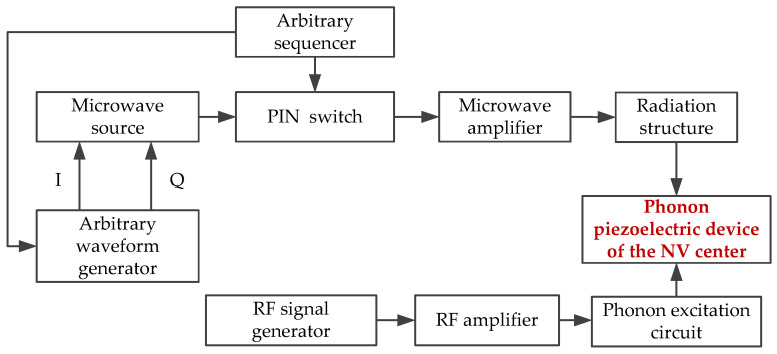
The architecture of the phonon-coupled manipulation module.

**Figure 21 micromachines-13-01628-f021:**
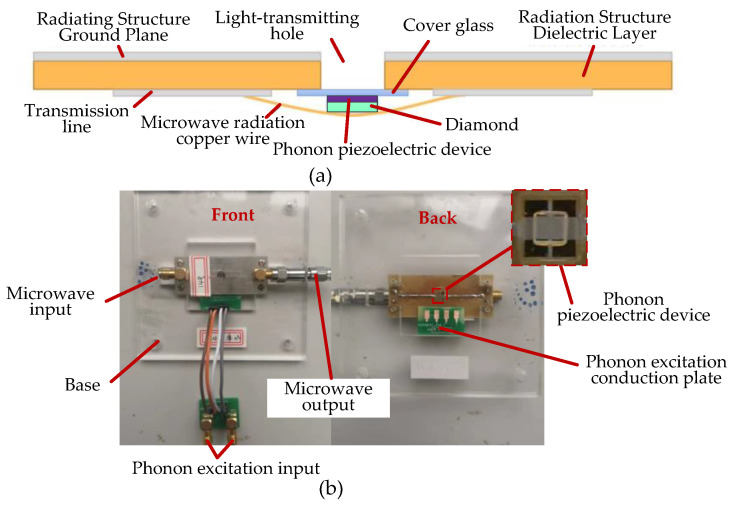
Phonon-coupled radiation structure and its physical photograph: (**a**) phonon-coupled radiation structure; (**b**) physical photograph.

**Figure 22 micromachines-13-01628-f022:**
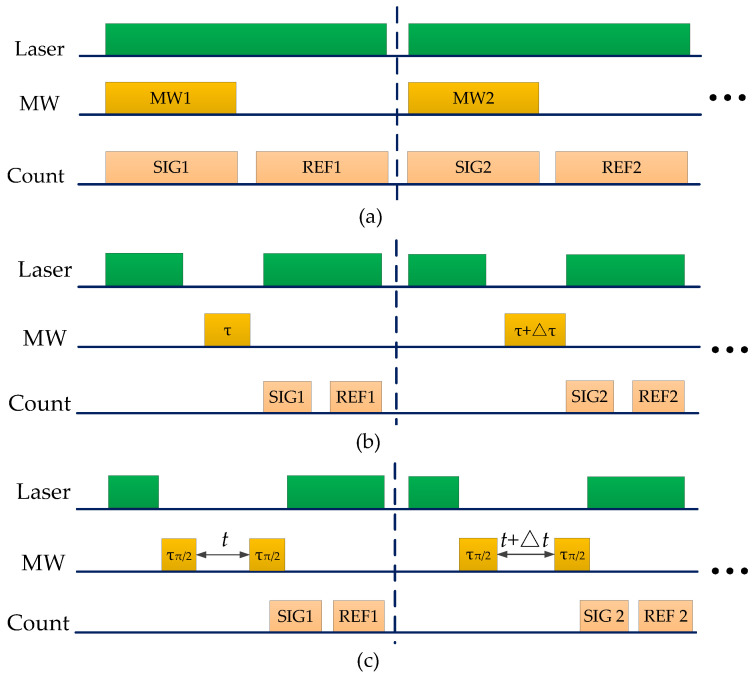
(**a**) ODMR measurement sequence. (**b**) Rabi measurement sequence. (**c**) Ramsey measurement sequence.

**Figure 23 micromachines-13-01628-f023:**
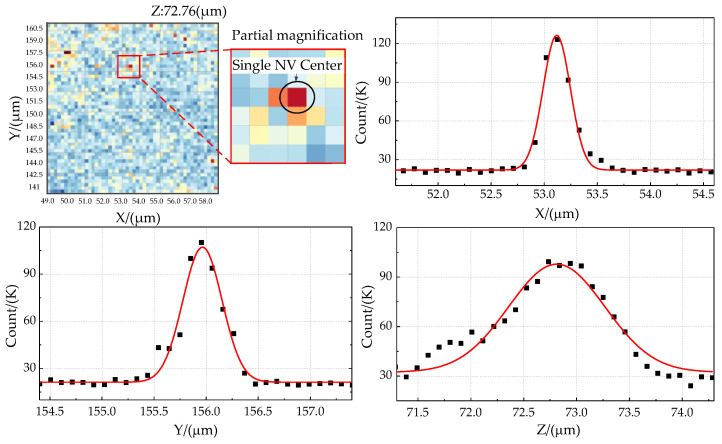
Characterization of the diamond sample with the NV center and its X, Y, and Z-axis fluorescence scanning curve.

**Figure 24 micromachines-13-01628-f024:**
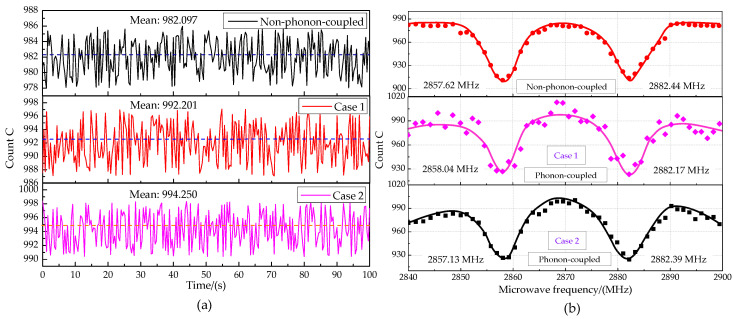
(**a**) Time-domain measurement at the MW frequency of 2870 MHz (**b**) ODMR spectrum.

**Figure 25 micromachines-13-01628-f025:**
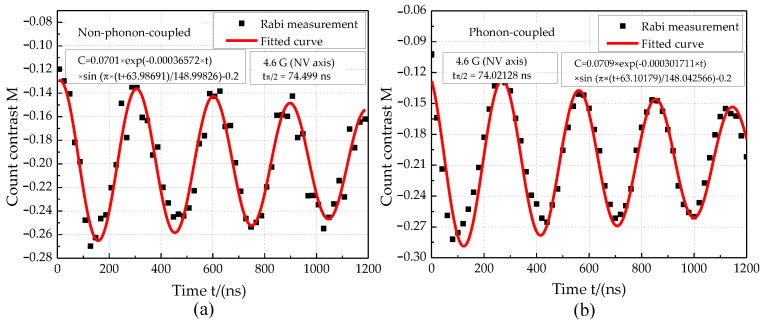
Rabi measurement results. (**a**) Non-phonon-coupled case. (**b**) Phonon-coupled case.

**Figure 26 micromachines-13-01628-f026:**
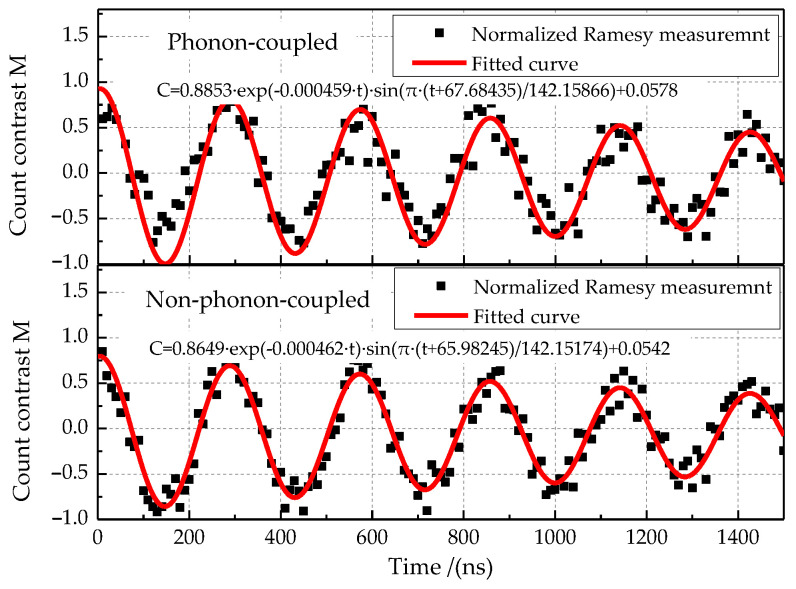
Ramsey measurement results.

**Table 1 micromachines-13-01628-t001:** Simulation structure parameters of the phonon piezoelectric device of the NV center.

Layer	Size	Value
IDT	Length/(μm)	10
	Width/(μm)	1
	Thickness/(μm)	0.1
	Spacing/(μm)	1
ZnO	Length/(μm)	70
	Width/(μm)	10
	Thickness/(μm)	0.4
Diamond	Length/(μm)	70
	Width/(μm)	10
	Thickness/(μm)	10

**Table 2 micromachines-13-01628-t002:** Material parameters of phonon piezoelectric device of the NV center.

Parameters	Symbol	Diamond	(100) ZnO	(002) ZnO
Elasticity coefficient (10^11^ N/m^2^)	*c* _11_	11.531	2.096	2.096
	*c* _12_	0.864	1.205	1.205
	*c* _13_	0.864	1.046	1.046
	*c* _33_	11.531	2.106	2.106
	*c* _44_	5.333	0.423	0.423
Temperature Coefficient (10^−4^/°C)	*Tc* _11_	−0.14	−1.12	−1.12
	*Tc* _12_	−0.57	−1.61	−1.61
	*Tc* _33_	−0.14	−1.23	−1.23
	*Tc* _44_	−0.125	−0.70	−0.70
Piezoelectric constant (C/m^2^)	*e* _15_	---	−0.48	—
	*e* _31_	---	−0.573	—
	*e* _33_	---	1.321	—
Relative permittivity	ε_11_/ε_0_	5.67	8.55	−0.48
	ε_33_/ε_0_	5.67	10.2	—
Density (10^3^ kg/m^3^)	*Ρ*	3.512	5.665	−0.573
Density Temperature Coefficient (10^−6^/°C)	*T*ρ	−3.6	−10.1	1.321

**Table 3 micromachines-13-01628-t003:** Optimized parameters of the IDT electrode size and the ZnO layer’s thickness.

Cases		The Thickness of the IDT/(nm)	Cases		The Thickness of the ZnO/(nm)
Case 1	a	80	Case 3	a	200
	b	100		b	400
	c	120		c	600
	d	140		d	800
Case 2	a	80	Case 4	a	200
	b	100		b	400
	c	120		c	600
	d	140		d	800

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
