# Peer review of "Structural Optimization and MEMS Implementation of the NV Center Phonon Piezoelectric Device"

_micromachines, 2022, doi:10.3390/mi13101628_

Round 1

Reviewer 1 Report (New Reviewer)

The authors design piezoelectric devices out of ZnO to mechanically manipulate the spins of NV centers in diamond based on thorough simulation and analysis. Below are some comments. 

  • Overall why I did not get why the authors chose ZnO. Please emphasize the reason, and hopefully with the pros and cons. 
  • In line 42, what does "the energy gap" mean? Energy of spin? 
  • In L89, E_y states are not defined or explained either. 
  • In Fig. 25, (a) and (b) must be flipped, or perhaps a typo in the caption.  

Author Response

Dear reviewer, thank you for your review and comments on the article - “Structural optimization and MEMS implement of the phonon piezoelectric device of the NV center”. The replies and modifications on your comments are listed one by one as follows. The authors hope that you can review them again during your busy schedule.

Point 1: Overall why I did not get why the authors chose ZnO. Please emphasize the reason, and hopefully with the pros and cons.

Response 1: The authors appreciate the reviewer’s suggestion. ZnO materials, as a kind of functional materials, with semiconductor properties, piezoelectric properties, good thermal stability and biocompatibility, have been widely used in the preparation of various functional devices. This material has important research value in the fields of photoelectricity, piezoelectricity, thermoelectricity, ferroelectricity and so on. As shown in the literature (Coupling a surface acoustic wave to an electron spin in diamond via a dark state. Phys. Rev. X 2016, 6: 041060.), the scholars also used the ZnO material as a surface acoustic wave piezoelectric material incentive to the NV center. The devices based on ZnO materials show a strong potential in the application of wearable electronic devices, flexible display, electronic skin, elf-powered sensing and so on. In this paper, the ZnO material can generate stable surface acoustic waves to assist control the transition of the NV center. Compared with general piezoelectric materials, ZnO material has higher price and greater difficulty in obtaining.

Point 2: In line 42, what does "the energy gap" mean? Energy of spin?

Response 2: The authors appreciate the reviewer’s suggestion. "The energy gap" means the difference between the lowest energies of the conduction band and the highest energies of the valence band of the NV center. The larger the energy gap of the NV center, the harder it is for electrons to be excited from the valence band to the conduction band, and the lower of the intrinsic carrier concentration. The deformation potential of the phonon-spin coupling strength of the NV center is related to the energy gap, and its spin coherence leads to a weak phonon coupling of the ground state spin.

Point 3: In L89, Ey states are not defined or explained either.

Response 3: The authors appreciate the reviewer’s suggestion. Ey is the strain-induced energy transfer excited state, which is defined in this paper.

Point 4: In Fig. 25, (a) and (b) must be flipped, or perhaps a typo in the caption.

Response 4: The authors appreciate the reviewer’s suggestion. Descriptions of (a) and (b) have been flipped in Fig. 25.

The above are the authors' responses. The authors appreciate the reviewer’s comments and suggestion again! Thank you very much!

Reviewer 2 Report (New Reviewer)

1. In the introduction, more recent reviews about the properties of NV centers should be added (e.g. A. Gali, Nanooptics 8, 1907 (2019))

2. At least once the authors should clarify that we are talking about a negatively charged NV-center

3. Section 2 is written unsatisfactorily.

a) The main equations are given without references, but they are not obtained by the authors. 

b) Exited states of NV-center are not defined (see Eq.1), Eq.2 is wriitem with a typo.

c) Subsec.2.2 and 2.3 are written without any references.

d) Line 164 has an incomprehensible numbering.

4. Sec.3 presents the main ideas, but equation (20) contains an error or typo. The presented relations have nothing to do with the boundary conditions. In this regard, the connection of the presented experimental results with the optimization conditions of the device is completely incomprehensible.

Author Response

Dear reviewer, thank you for your review and comments on the article - “Structural optimization and MEMS implement of the phonon piezoelectric device of the NV center”. The replies and modifications on your comments are listed one by one as follows. The authors hope that you can review them again during your busy schedule.

Point 1: In the introduction, more recent reviews about the properties of NV centers should be added (e.g. A. Gali, Nanooptics 8, 1907 (2019))

Response 1: The authors appreciate the reviewer’s suggestion. More recent reviews about the properties of NV centers (e.g. Ádám Gali. Ab initio theory of the nitrogen-vacancy center in diamond[J]. Nanophotonics, 2019, 8: 1907) have been added in this paper.

Point 2: At least once the authors should clarify that we are talking about a negatively charged NV-center.

Response 2: The authors appreciate the reviewer’s suggestion. A negatively charged NV center is selected in this paper, which contains six electrons, two from nitrogen atoms, three from carbon atoms adjacent to the vacancy, and one trapped (from donor impurity) electron. The nitrogen nucleus of the NV center contains five positively charged protons. Therefore, NV- center presents a negative one valence. Relevant descriptions have been described in the paper.

Point 3: In Section 2. a) The main equations are given without references, but they are not obtained by the authors. b) Exited states of NV-center are not defined (see Eq.1), Eq.2 is written with a typo. c) Subsec.2.2 and 2.3 are written without any references. d) Line 164 has an incomprehensible numbering.

Response 3: The authors appreciate the reviewer’s suggestion. The main equations in Section 2 have been given with references. Ey is the strain-induced energy transfer excited state, which is defined in this paper. References are added in Subsec.2.2 and 2.3. In Line 164, the incomprehensible numbering has been changed, where  is the reflection coefficient of the phonon device.

Point 4: Sec.3 presents the main ideas, but equation (20) contains an error or typo. The presented relations have nothing to do with the boundary conditions. In this regard, the connection of the presented experimental results with the optimization conditions of the device is completely incomprehensible.

Response 4: The authors appreciate the reviewer’s suggestion. Equation (20) has been corrected in this paper. Sec.3 presents the finite element model and boundary conditions of the phonon piezoelectric device. The phase velocity v and the electromechanical coupling coefficient K2 are analyzed in this paper, which is corresponds to Section 2.2. The phase velocity determines the center frequency and delay time of the phonon piezoelectric device. The electromechanical coupling coefficient determines the acoustic-electric conversion efficiency of the phonon piezoelectric device. On the basis, the structure of the phonon piezoelectric device of the NV center is optimized, and its MEMS implementation and characterization are carried out. Experimental results show that the phonon resonance manipulation method can effectively increase the NV center's spin transition probability based on the MEMS phonon piezoelectric device prepared in this paper, improving quantum spin manipulation efficiency.

The above are the authors' responses. The authors appreciate the reviewer’s comments and suggestion again! Thank you very much!

Reviewer 3 Report (New Reviewer)

The authors designed and demonstrated a phonon piezoelectric device for the NV centers to improve the spin transition probability for higher quantum spin manipulation efficiency. The topic is very interesting and important for the quantum sensing based on NV centers in diamond. The manuscript is well organized and written with a lot of details. I would suggest the manuscript to be accepted by Micromachines.

Author Response

Dear reviewer, thank you for your review and comments on the article - “Structural optimization and MEMS implement of the phonon piezoelectric device of the NV center”. The replies and modifications on your comments are listed one by one as follows. The authors hope that you can review them again during your busy schedule.

Point 1: The authors designed and demonstrated a phonon piezoelectric device for the NV centers to improve the spin transition probability for higher quantum spin manipulation efficiency. The topic is very interesting and important for the quantum sensing based on NV centers in diamond. The manuscript is well organized and written with a lot of details. I would suggest the manuscript to be accepted by Micromachines.

Response 1: Thanks to the reviewer, some revisions has done to the newest manuscript.

The above are the authors' responses. The authors appreciate the reviewer’s comments and suggestion again! Thank you very much!

Round 2

Reviewer 2 Report (New Reviewer)

The manuscript is now better after revision. However, there are still comments, including significant ones.

1. The basic equations (20)-(21) should either be corrected or explained in more detail. 

Namely, the first equation in the system (20) is independent of the second, which is just an algebraic relation. However, the authors solve systems of differential equations. Therefore, further conclusions have no justification.

2. If the system of equations (20) is original and belongs to the authors, it should be strictly explained, if not, there should be a corresponding reference.

3. Since the displacement u is a vector, this should be indicated in equations (20). Since 2 vectors (displacement and angle) are used in the same equation, their equal dimension must be justified.

4. Typos are not eliminated everywhere, see, for example, Eq. (2).

Author Response

Dear reviewer, thank you for your review and comments on the article - “Structural optimization and MEMS implement of the phonon piezoelectric device of the NV center”. The replies and modifications on your comments are listed one by one as follows. The authors hope that you can review them again during your busy schedule.

Point 1-3: The basic equations (20)-(21) should either be corrected or explained in more detail, Namely, the first equation in the system (20) is independent of the second, which is just an algebraic relation. However, the authors solve systems of differential equations. Therefore, further conclusions have no justification; If the system of equations (20) is original and belongs to the authors, it should be strictly explained, if not, there should be a corresponding reference; Since the displacement u is a vector, this should be indicated in equations (20). Since 2 vectors (displacement and angle) are used in the same equation, their equal dimension must be justified.

Response 1-3: The authors appreciate the reviewer’s suggestion. Since points 1-3 are questions of the same object, they are answered together here. Finite element model of the phonon piezoelectric device is established in this paper and no references (Response 2). The model is modified and elaborated as follows (Response 1):

The matrix equation of the piezoelectric coupling field of the phonon piezoelectric device of the NV center is expressed as:

 (20)

where M is the mass matrix of the ZnO phonon piezoelectric membrane;  is the mechanical body force matrix;  is the body force matrix;  is the mechanical surface force matrix;  is the mechanical point force matrix;  is the mechanical damping matrix;  is the piezoelectric coupling matrix;  is the dielectric stiffness matrix;  is the surface charge matrix;  is the point charge matrix;  is the nodal displacement in natural mode for  time.  is the electric potential for  time. (Response 3: Both of u(t) and (t) are time-varying first-order equation. Both of  and  are specific value, and have the same dimensions.)

Regardless of the mechanical damping (let ), and  is eliminated. The matrix equation of the ZnO piezoelectric membrane of phonon piezoelectric device is:

 (21)

The homogeneous solution of Eq. (21) corresponds to the various modes in the model and the frequencies corresponding to the various modes. The standard solution of u(t) is:

 (22)

where u is the nodal displacement matrix in natural mode.  is the modal frequency. According to Eq. (21) and Eq. (22), it can obtained that:

(23)

Its standard solution satisfies:

 (24)

The modal frequency  can be solved by Eq.(24). Its modal displacement  can be solved by Eq. (23).

Relevant descriptions have been modified in the paper.

 Point 4. Typos are not eliminated everywhere, see, for example, Eq. (2).

Response 4: The authors appreciate the reviewer’s suggestion. Typos have been checked again. The Eq. (2) refer to the literature [35] (Coupling a surface acoustic wave to an electron spin in diamond via a dark state. Phys. Rev. X 2016, 6: 041060.), and typos are corrected. It is shown as follows:

(2).

The above are the authors' responses. The authors appreciate the reviewer’s comments and suggestion again! Thank you very much!

Round 3

Reviewer 2 Report (New Reviewer)

It is easy to see that the first equation in the system (20) does not depend on the second. The first equation is differential of the second order, and the second is algebraic. Therefore, the solution of the second is determined by the solution of the first. The first equation does not depend on the function phi at all. Please resolve this misunderstanding.

Author Response

Dear reviewer, thank you for your review and comments on the article - “Structural optimization and MEMS implement of the phonon piezoelectric device of the NV center”. The replies and modifications on your comments are listed one by one as follows. The authors hope that you can review them again during your busy schedule.

Point 1: It is easy to see that the first equation in the system (20) does not depend on the second. The first equation is differential of the second order, and the second is algebraic. Therefore, the solution of the second is determined by the solution of the first. The first equation does not depend on the function phi at all. Please resolve this misunderstanding.

Response 1: The authors appreciate the reviewer’s suggestion. This expression before misunderstands readers. Eq. (20) is not an equation set, but also two independent equations. The first equation of Eq. (20) is the differential of the second order of the phonon piezoelectric device of the NV center. The second equation of Eq. (20) is the matrix equation of the piezoelectric coupling field of the phonon piezoelectric device of the NV center. These two equations describe the mechanical and piezoelectric characteristics of the device, respectively. In order not to misunderstand the readers, relevant descriptions have been modified accord to reviewer shown as follows:

The mechanical characteristic of the phonon piezoelectric device of the NV center is expressed as:

(20)

The matrix equation of the piezoelectric coupling field of the phonon piezoelectric device of the NV center is :

(21)

In Eq. (20) and Eq. (21), M is the mass matrix of the ZnO phonon piezoelectric membrane;  is the mechanical body force matrix;  is the body force matrix;  is the mechanical surface force matrix;  is the mechanical point force matrix;  is the mechanical damping matrix;  is the piezoelectric coupling matrix;  is the surface charge matrix;  is the point charge matrix;  is the nodal displacement in natural mode for  time.  is the electric potential for  time.

Relevant descriptions have been modified in the paper.

The above are the authors' responses. The authors appreciate the reviewer’s comments and suggestion again! Thank you very much!

This manuscript is a resubmission of an earlier submission. The following is a list of the peer review reports and author responses from that submission.

Round 1

Reviewer 1 Report

The manuscript by Shen et al. describes the basic theory, design, simulation, fabrication, and implementation of surface acoustic wave (SAW) devices on a diamond surface, with the aim of phonon-assisted control of the single electronic spin of the NV center in diamond. I do not think the quality of the present manuscript is particularly high or novel, but the topic is of interest to the NV-center research community and the description of the manuscript covering from basic theory through implementation will be helpful to get the overview of this topic. It is thus in principle publishable in the present journal. However, the most important part, experimental demonstration, is very poorly presented. First of all, the authors explained in Fig. 1(b) that the mechanism of the phonon-assisted spin control is that a red-detuned optical pulse couples with the phononic field to provide an effective spin transition. The experimental sequence in Fig. 22, however, continuously illuminates the green laser for readout, which is much shorter than the transition wavelength (known not to cause the spin transition like Fig. 1(b) No phonon field in the first place). Even if we trust the analysis based on Fig. 24, a-few-percent improvement after very laborious and complicated fabrication procedures is not really worthwhile. More experimental data are necessary to justify this research program: for instance, time-domain measurement. Also, the strain control could permit magnetically-forbidden between m_S = +1 and -1 states, as demonstrated previously (e.g., Ref [27]). Finally, English has to be polished. At the very least, the title should be revised.

Author Response

Dear reviewer, thank you for your review and comments on the article - “Structural optimization and MEMS implement of the phonon piezoelectric device of the NV center”. The replies and modifications on your comments are listed one by one as follows. The authors hope that you can review them again during your busy schedule.

(1) Point 1: The experimental demonstration is poorly presented.

Response 1: The authors appreciate the reviewer’s suggestion. The experimental demonstration has been improved. Especially, the time-domain measurement is added in this paper. Time-domain measurement at the MW frequency of 2870 MHz and the ODMR spectrum for different cases are shown in Figure 24, including the non-phonon-coupled case and the phonon-coupled case (including Case 1 and Case 2). In Case 1, IDT is deposited on the surface of the (100)ZnO piezoelectric membrane layer. In Case 2, IDT is deposited on the interface of the diamond and (100)ZnO piezoelectric membrane. Time-domain measurement result shows that phonon-coupled case with the average fluorescence count at the MW frequency of 2870 MHz are about 987 within 100 s. That count for non-phonon-coupled case in the same condition is about 982. It can be shown from the ODMR spectrum that the  state is split into  and . The microwave resonance frequencies for the non-phonon-coupled case are 2857.62 MHz and 2882.44 MHz. The microwave resonance frequencies for the phonon-coupled case are about 2857 MHz and 2882 MHz. Both of which are symmetrical about 2870 MHz. However, the fluorescence intensity for the phonon-coupled case is higher than that of the non-phonon-coupled case. In Case 1 and Case 2, the average fluorescence intensity increased by 1.02% and 1.29% compared with non-phonon-coupled case, respectively. The possible reason is that more photons jump with phonon-coupled manipulation. This phenomenon is more obvious around the microwave frequencies. The bandwidth of the resonance peak for the phonon-coupled case is larger. Compared to the Case 2, the fluorescence intensity is more unstable for Case 1.

(2) Point 2: The authors explained in Fig. 1(b) that the mechanism of the phonon-assisted spin control is that a red-detuned optical pulse couples with the phononic field to provide an effective spin transition. The experimental sequence in Fig. 22, however, continuously illuminates the green laser for readout, which is much shorter than the transition wavelength (known not to cause the spin transition like Fig. 1(b) No phonon field in the first place).

Response 2: The authors appreciate the reviewer’s comment. As reviewer understand, with the phonon-coupled manipulation method, multi-physics fields such as magnetic field, radiofrequency (RF) field, microwave, and laser perform resonant coupling regulation of the NV center. The optical-driven spin transition system is within the large-dipole detuning limit. The spin transition is manipulated by the phonon coupling of the phonon field-driven system from "|g>" state to "|e>" state in the presence of a phonon-containing field. This system is equivalent to a spin transition between two lower-state phonon ladders.

The NV center can be divided into the negative monovalent NV center (NV-), the zero-valent NV center (NV0) and the positive monovalent NV center(NV+). The NV center in this paper is the NV-. The free electrons in the NV- center structure can be subjected to spin polarization and transition manipulation by a 532 nm laser(green), and detected by the released fluorescence signal, and the signal to be measured can be obtained after calculation. The zero-phonon line of the NV center is 637 nm (red). However, as you know, the experimental sequence in Fig. 22 continuously illuminates the green laser for readout, which is much shorter than the transition wavelength. The reason is that the frequency of the laser is not a single value, but is in a normal distribution interval, and the set laser frequency is the middle value of the normal distribution. For the non-phonon-coupled case, photons with wavelengths of 532 nm (green, whose corresponding actual Larmor frequency is  ) can achieve excitation of the NV center. For the phonon-coupled case, photons with wavelengths less than 532nm (close to green, whose corresponding actual Larmor frequency is ) can achieve excitation of the NV center. More photons transition from the ground state to the excited state for the phonon-coupled case. So, it can improve the quantum spin manipulation efficiency of the NV center. Moreover, phonons of different frequencies have different effects on the auxiliary transition of the NV center for a fixed frequency laser. So, we need to optimize the phonon device structure to generate phonons of different frequencies to further improve the transition efficiency of the NV center.

(3) Point 3: In Fig. 24, a-few-percent improvement after very laborious and complicated fabrication procedures is not really worthwhile. More experimental data are necessary to justify this research program: for instance, time-domain measurement.

Response 3: The authors appreciate the reviewer’s suggestion. As shown in the literature (Atomic spin and phonon coupling mechanism of nitrogen-vacancy center. Acta Phys. Sin. 2021, 70(6): 068501.) opened by our team before, the main resonance frequency of piezoelectric phonon device of the NV center is in the frequency magnitude of THz, and it has a better coupling control effect at this phonon excitation frequency. Due to the limitation of the current MEMS fabrication capability of the IDT of piezoelectric phonon device of the NV center, the width of the IDT in this paper is 3 μm, and the corresponding phonon excitation frequency is 350 MHz, which is in the sub-resonant frequency range. The phonon-coupled parameters are still not optimal. The calculation method of the phonon excitation frequency is as follows: 350 MHz. where  is the propagation speed of phonon wave.  is the IDT width. It is worth noting that phonon-coupled NV center with phonon excitation frequency of THz has a high phonon density of states, and can achieve high transition efficiency improvements theoretically. The realization of the phonon excitation frequency of 350 MHz in this paper is to verify the feasibility of the phonon coupling manipulation effect. In addition, more experimental data such as the time-domain measurement is added in this paper to further verify the feasibility of the phonon coupling manipulation effect. Description of relevant results are shown in Point 1.

(4) Point 4: The strain control could permit magnetically-forbidden between mS = +1 and -1 states, as demonstrated previously (e.g., Ref [27]).

Response 4: The authors appreciate the reviewer’s suggestion. Inhomogeneous dephasing from uncontrolled environmental noise can limit the coherence of a quantum sensor or qubit. In Ref [27], authors prolonged the dephasing time by dressing with mechanical Rabi field. Furthermore, they developed a model that quantitatively predicts the relationship between  and the dephasing time in the dressed basis. The magnetic field fluctuation as a disturbance term needs to be suppressed or compensated by the mechanically driven field. The role of the intrinsic magnetic field in the NV center transition is to cause Zeeman splitting between the mS = +1 and mS = -1 states. So, the intrinsic magnetic field does not need to be forbidden in this process, but to suppress or compensate the magnetic field disturbance. In the experiment of our paper, we choose a permanent magnet of 80 G applied along the NV axis, combined with the magnetic shielding method, achieving a relatively constant of the magnetic field.

(5) Point 5: English has to be polished. At the very least, the title should be revised.

Response 5: The authors appreciate the reviewer’s suggestion. English expression has been checked and improved. Especially, the title has been revised to “Structural optimization and MEMS implement of the phonon piezoelectric device of the NV center”.

In addition to the above modifications, additional modifications are performed. The main contents are listed as follows:

Point 1: Explanations of the results of structural optimization have been added to this paper. The phase velocity of the phonon piezoelectric device of the NV center is related to the center frequency and the delay time. The electromechanical coupling coefficient of the phonon piezoelectric device defined as the ratio of acoustic wave mechanical energy to excitation electrical energy, which can measure the acoustoelectric conversion efficiency. Higher phase velocity and higher electromechanical coupling coefficient has better electromechanical coupling performance, so as to obtain better NV center transition efficiency. In this paper, Fig. 12 and Fig. 15 characterize the above two characteristics. Simulation results show that the electromechanical coupling coefficient K2 of (100)ZnO phonon device is larger than that of (002)ZnO phonon device. When the thickness of the (100)ZnO piezoelectric membrane layer is [400, 600] nm, it has optimized electromechanical coupling coefficients for four modes. This paper only simulates the IDT space of 1 μm, 2 μm and 3 μm, the phonon state density  for different IDT space is: . Since the IDT spacing determines the frequency of the phonon, it had been analyzed in the literature opened by our team before. The main resonance frequency of piezoelectric phonon device of the NV center is in the frequency magnitude of THz, and it has a better coupling control effect at this phonon excitation frequency. Relevant structural optimization results have been added to this paper.

Point 2: We did not explain the significance of these formulas for practical application before, such as (15) - (19). The relevant description has been supplemented. The differential calculation method of phonon piezoelectric device of the NV center is described from Eq. (15) to Eq. (19). They describe the IDT structure model parameters of the phonon piezoelectric device in Fig. 2. They provide theoretical support for the structural optimization of phonon piezoelectric devices of the NV center. In Eq. (15) and Eq. (16), the characteristic function of the phonon field is elaborated. The differential decomposition equation of the phonon piezoelectric device is described in Eq. (19). Based on Eq. (15) to Eq. (19), we can construct the model of the IDT, providing model and parameter support for structural optimization of phonon piezoelectric device of the NV center and phonon-coupled manipulation experiment of the NV center. In Eq. (15) to Eq. (19), specific parameters are explained. where β is the acoustic wave number; q is the difference between the wave number and the crystal wave number;  is the acoustic wave field in the +x direction;  is the acoustic wave field in the -x direction.  and  are gradient field in space . where  is the acoustic wave velocity;  is the acoustic wave loss. where  is the reflection coefficient of the phonon device;  is the transduction coefficient of the interdigital transducer;  is the capacitance;  is the excitation amplitude voltage. Relevant descriptions for Eq. (15) to Eq. (19) have been supplemented in this paper.

The above are the authors' responses. The authors appreciate the reviewer’s comments and suggestion again! Thank you very much!

Reviewer 2 Report

In this manuscript, the structure of phonon-coupled manipulation piezoelectric device of the NV center is designed and optimized. However, it seems not to provide the experiments toward the structural optimization.

1.      The optimized parameters are focused on the IDT electrode size and the ZnO layers thickness. How about other dimension parameters? Or how are the other parameters determined?

2.      Can the structural optimization for the comparison with the simulations and  experiments?

3.      There seems to be lacking explanations of the results of structural optimization.

4.      What are the advantages of this work, especially compare with other phonon piezoelectric devices?

5.      This paper involves many formula calculations, such as (15) - (19). The author does not explain the significance of these formulas for practical application. It is suggested to supplement them.

6.      Figure 9 shows phonon piezoelectric device structure of the NV center for 4 cases, Which one is the last one and why?

Author Response

Dear reviewer, thank you for your review and comments on the article - “Structural optimization and MEMS implement of the phonon piezoelectric device of the NV center”. The replies and modifications on your comments are listed one by one as follows. The authors hope that you can review them again during your busy schedule.

(1) Point 1: The optimized parameters are focused on the IDT electrode size and the ZnO layer’s thickness. How about other dimension parameters? Or how are the other parameters determined?

Response 1: The authors appreciate the reviewer’s question. In the transition process of the phonon-assisted NV center, the phonon vibration frequency and phonon vibration amplitude of the diamond lattice are key parameters affecting the transition efficiency of the NV center, which are determined by the IDT excitation parameters and the thickness of the ZnO piezoelectric layer. So, it is necessary to optimize those parameters. Other parameters such as the plane size of ZnO piezoelectric layer and the size of the diamond have little effect on the transition efficiency of the phonon-assisted NV center. It is laborious and insignificant to optimizing this kind of parameters. In this paper, we refer to Ref [34] for the selection of parameters. In addition, the NV centers inside the diamond are widely distributed. So, the size parameters of diamond do not affect the selection of the single NV center. In summary, it is sufficient to optimize the IDT excitation structure parameters and piezoelectric material thickness in this paper with reference to the work of predecessors.

(2) Point 2: Can the structural optimization for the comparison with the simulations and experiments?

Response 2: The authors appreciate the reviewer’s question. As shown in the literature (Atomic spin and phonon coupling mechanism of nitrogen-vacancy center. Acta Phys. Sin. 2021, 70(6): 068501.) opened by our team before, the main resonance frequency of piezoelectric phonon device of the NV center is in the frequency magnitude of THz, and it has a better coupling control effect at this phonon excitation frequency. Due to the limitation of the current MEMS fabrication capability of the IDT of piezoelectric phonon device of the NV center, the width of the IDT in this paper is 3 μm, and the corresponding phonon excitation frequency is 350 MHz, which is in the sub-resonant frequency range. The phonon-coupled parameters are still not optimal. The calculation method of the phonon excitation frequency is as follows: 350 MHz. where  is the propagation speed of phonon wave.  is the IDT width. It is worth noting that phonon-coupled NV center with phonon excitation frequency of THz has a high phonon density of states, and can achieve high transition efficiency improvements theoretically. The realization of the phonon excitation frequency of 350 MHz in this paper is to verify the feasibility of the phonon coupling manipulation effect. In addition, under the simulation conditions of this paper, when the thickness of the (100)ZnO piezoelectric membrane layer is [400, 600] nm, it has optimized electromechanical coupling coefficients for four modes. Since the sample was prepared to an optimal size in the simulation condition in this paper, Case 1 in simulation (IDT is deposited on the surface of the piezoelectric membrane layer) and Case 2 in simulation (IDT is deposited on the interface of the diamond and piezoelectric membrane) are verified experimentally in this paper. The sizes of the phonon piezoelectric device are described in Fig.16.

(3) Point 3: There seems to be lacking explanations of the results of structural optimization.

Response 3: The authors appreciate the reviewer’s suggestion. The author has added relevant explanations. The phase velocity of the phonon piezoelectric device of the NV center is related to the center frequency and the delay time. The electromechanical coupling coefficient of the phonon piezoelectric device defined as the ratio of acoustic wave mechanical energy to excitation electrical energy, which can measure the acoustoelectric conversion efficiency. Higher phase velocity and higher electromechanical coupling coefficient has better electromechanical coupling performance, so as to obtain better NV center transition efficiency. In this paper, Fig. 12 and Fig. 15 characterize the above two characteristics. Simulation results show that the electromechanical coupling coefficient K2 of (100)ZnO phonon device is larger than that of (002)ZnO phonon device. When the thickness of the (100)ZnO piezoelectric membrane layer is [400, 600] nm, it has optimized electromechanical coupling coefficients for four modes. This paper only simulates the IDT space of 1 μm, 2 μm and 3 μm, the phonon state density  for different IDT space is: . Since the IDT spacing determines the frequency of the phonon, it had been analyzed in the literature opened by our team before. The main resonance frequency of piezoelectric phonon device of the NV center is in the frequency magnitude of THz, and it has a better coupling control effect at this phonon excitation frequency. Relevant structural optimization results have been added to this paper.

(4) Point 4: What are the advantages of this work, especially compare with other phonon piezoelectric devices?

Response 4: The authors appreciate the reviewer’s question. Although the theoretical research on phonon-spin interaction has been preliminary carried out in recent years, the phonon field excitation module (phonon piezoelectric device) coupled with the radio frequency field, microwave field and magnetic field has not been studied. Also, the electromechanical optimization problem of the phonon piezoelectric device has not been solved before. This work is the first time to optimize the electromechanical coupling performance of phonon piezoelectric device of the NV center. The purpose is to improve the transition efficiency of the NV center, and this work is groundbreaking.

(5) Point 5: This paper involves many formula calculations, such as (15) - (19). The author does not explain the significance of these formulas for practical application. It is suggested to supplement them.

Response 5: The authors appreciate the reviewer’s suggestion. The relevant description has been supplemented. The differential calculation method of phonon piezoelectric device of the NV center is described from Eq. (15) to Eq. (19). They describe the IDT structure model parameters of the phonon piezoelectric device in Fig. 2. They provide theoretical support for the structural optimization of phonon piezoelectric devices of the NV center. In Eq. (15) and Eq. (16), the characteristic function of the phonon field is elaborated. The differential decomposition equation of the phonon piezoelectric device is described in Eq. (19). Based on Eq. (15) to Eq. (19), we can construct the model of the IDT, providing model and parameter support for structural optimization of phonon piezoelectric device of the NV center and phonon-coupled manipulation experiment of the NV center. In Eq. (15) to Eq. (19), specific parameters are explained. where β is the acoustic wave number; q is the difference between the wave number and the crystal wave number;  is the acoustic wave field in the +x direction;  is the acoustic wave field in the -x direction.  and  are gradient field in space . where  is the acoustic wave velocity;  is the acoustic wave loss. where  is the reflection coefficient of the phonon device;  is the transduction coefficient of the interdigital transducer;  is the capacitance;  is the excitation amplitude voltage. Relevant descriptions for Eq. (15) to Eq. (19) have been supplemented in this paper.

(6) Point 6: Figure 9 shows phonon piezoelectric device structure of the NV center for 4 cases, Which one is the last one and why?

Response 6: The authors appreciate the reviewer’s question. As is described in Point 2, the main resonance frequency of piezoelectric phonon device of the NV center is in the frequency magnitude of THz, and it has a better coupling control effect at this phonon excitation frequency. Due to the limitation of the current MEMS fabrication capability of the IDT of piezoelectric phonon device of the NV center, the width of the IDT in this paper is 3 μm, and the corresponding phonon excitation frequency is 350 MHz, which is in the sub-resonant frequency range. The phonon-coupled parameters are still not optimal. It is worth noting that phonon-coupled NV center with phonon excitation frequency of THz has a high phonon density of states, and can achieve high transition efficiency improvements theoretically. The realization of the phonon excitation frequency of 350 MHz in this paper is to verify the feasibility of the phonon coupling manipulation effect. In addition, under the simulation conditions of this paper, when the thickness of the (100)ZnO piezoelectric membrane layer is [400, 600] nm, it has optimized electromechanical coupling coefficients for four modes. So, The last size of the the phonon piezoelectric device are described in Fig.16. They are optimized structure size parameters for the conditions that our experiments can achieve, and corresponds to the simulation.

In addition to the above modifications, additional modifications are performed. The main contents are listed as follows:

Point 1: Optimized the expression of experimental demonstration. The experimental demonstration has been improved. Especially, the time-domain measurement is added in this paper. Time-domain measurement at the MW frequency of 2870 MHz and the ODMR spectrum for different cases are shown in Figure 24, including the non-phonon-coupled case and the phonon-coupled case (including Case 1 and Case 2). In Case 1, IDT is deposited on the surface of the (100)ZnO piezoelectric membrane layer. In Case 2, IDT is deposited on the interface of the diamond and (100)ZnO piezoelectric membrane. Time-domain measurement result shows that phonon-coupled case with the average fluorescence count at the MW frequency of 2870 MHz are about 987 within 100 s. That count for non-phonon-coupled case in the same condition is about 982. It can be shown from the ODMR spectrum that the  state is split into  and . The microwave resonance frequencies for the non-phonon-coupled case are 2857.62 MHz and 2882.44 MHz. The microwave resonance frequencies for the phonon-coupled case are about 2857 MHz and 2882 MHz. Both of which are symmetrical about 2870 MHz. However, the fluorescence intensity for the phonon-coupled case is higher than that of the non-phonon-coupled case. In Case 1 and Case 2, the average fluorescence intensity increased by 1.02% and 1.29% compared with non-phonon-coupled case, respectively. The possible reason is that more photons jump with phonon-coupled manipulation. This phenomenon is more obvious around the microwave frequencies. The bandwidth of the resonance peak for the phonon-coupled case is larger. Compared to the Case 2, the fluorescence intensity is more unstable for Case 1.

Point 2: English expression has been checked and improved. Especially, the title has been revised to “Structural optimization and MEMS implement of the phonon piezoelectric device of the NV center”.

The above are the authors' responses. The authors appreciate the reviewer’s comments and suggestion again! Thank you very much!

Round 2

Reviewer 1 Report

The revision is unsatisfactory.

Regarding Response 1, the authors misunderstood what is meant by ''time-domain measurement.'' In the broad quantum technology community (including the NV-center research community), it is unambiguously synonymous with ''pulsed measurement,'' such as Ramsey interferometry and spin echo, which can reveal coherence properties of the NV-center spin that are missing in the present work. I do not see any new useful information from the added data in Fig. 24(a). Instead of Fig. 24(a), the authors can add the error bars to each data point in Fig. 24(b), so that the noise (fluctuations of the photon counts) in the presence/absence of the phonon field can be evaluated in the entire frequency range. But again, this does not reveal coherence properties of the spin.

Response 2 is incorrect. The optical transition by the 532-nm laser and the subsequent photon emission are spin-preserving. The spin-flip mechanism responsible for the initialization of the NV-center spin arises from the non-radiative transition through intersystem crossing mediated by the spin-singlet states. The mechanism described in Fig. 1(b) cannot be applied in the case of the optical transition by the 532-nm laser.

I do not think Response 3 answers my comment. If the THz phonon field is required to achieve high efficiency and the current fabrication technology is no way close to achieving this, I do not see any benefits of testing at 350 MHz, which is four orders of magnitude slower that needed.

In Response 4, the authors misinterpreted my comment. I was asking whether the photon field generated by SAW could directly drive the -1 to/from +1 transition via the modification of the strain term of the NV-spin Hamiltonian.

Author Response

Relevant responses are listed in the attachment, because of some formulas and figures cannot be displayed here.

Reviewer 2 Report

No required change, it can be accepted.

Author Response

Thanks to the reviewer, some revisions has done to the newest manuscript. We marked them in the paper. Thank you very much!